# Forecasting individual progression trajectories in Alzheimer's disease

Etienne Maheux [1], Igor Koval[1], Juliette Ortholand[1], Colin Birkenbihl[2,3], Damiano Archetti [4], Vincent Bouteloup[5,6], Stéphane Epelbaum[7], Carole Dufouil [5,6], Martin Hofmann-Apitius [2,3] & Stanley Durrleman [1] ✉

The anticipation of progression of Alzheimer's disease (AD) is crucial for evaluations of secondary prevention measures thought to modify the disease trajectory. However, it is difficult to forecast the natural progression of AD, notably because several functions decline at different ages and different rates in different patients. We evaluate here AD Course Map, a statistical model predicting the progression of neuropsychological assessments and imaging biomarkers for a patient from current medical and radiological data at early disease stages. We tested the method on more than 96,000 cases, with a pool of more than 4,600 patients from four continents. We measured the accuracy of the method for selecting participants displaying a progression of clinical endpoints during a hypothetical trial. We show that enriching the population with the predicted progressors decreases the required sample size by 38% to 50%, depending on trial duration, outcome, and targeted disease stage, from asymptomatic individuals at risk of AD to subjects with early and mild AD. We show that the method introduces no biases regarding sex or geographic locations and is robust to missing data. It performs best at the earliest stages of disease and is therefore highly suitable for use in prevention trials.

The cost of drug development is highest, by far, for neurodegenerative diseases, with unparalleled failure rates[1]. In this respect, the controversial approval of aducanumab on 7 June 2021 by the Food and Drug Administration (FDA) represents a turning point in Alzheimer's disease (AD) drug development[2]. This decision raises the critical issue of demonstrating the clinical benefit of a compound acting on a key biological process, the accumulation of amyloid plaques in the brain[3].

It remains unclear why an effective intervention for such a key biological mechanism is only weakly associated with lower levels of cognitive decline. It is likely that the core biological processes and their interactions are not yet fully understood. Another, non-exclusive

explanation is that the issue of who and when to treat must be addressed with greater precision to demonstrate clinical efficacy. In 2019, Cummings and coworkers were already stressing the need to improve clinical trials, by targeting the right participant with the right biomarker in the right trial[4]. The motivation, here, is simple: it is not possible to show that a candidate therapy slows down the degradation of the endpoint if this endpoint is not expected to worsen during the trial. The treatment effect size will be larger if one includes participants right before the disease progression would cause a significant change in the endpoint without an intervention. Such a target period depends on the endpoint selected to demonstrate efficacy.

[1]Sorbonne Université, Institut du Cerveau - Paris Brain Institute – ICM, CNRS, Inria, Inserm, AP-HP, Hôpital Pitié-Salpêtrière, Paris, France. [2]Department of bioinformatics, Fraunhofer Institute for Algorithms and Scientific Computing (SCAI), Sankt Augustin, Germany. [3]Bonn-Aachen International Center for IT, Rheinische Friedrich-Wilhelms-Universität Bonn, Bonn 53115, Germany. [4]IRCCS Instituto Centro San Giovanni di Dio Fatebenefratelli, Brescia, Italy. [5]Université de Bordeaux, CNRS UMR 5293, Institut des Maladies Neurodégénératives, Bordeaux, France. [6]Centre Hospitalier Universitaire (CHU) de Bordeaux, pôle de neurosciences cliniques, centre mémoire de ressources et de recherche, Bordeaux, France. [7]Sorbonne Université, Institut du Cerveau - Paris Brain Institute – ICM, CNRS, Inria, Inserm, AP-HP, Hôpital Pitié-Salpêtrière, Institut de la mémoire et de la maladie d'Alzheimer (IM2A), center of excellence of neurodegenerative diseases (CoEN), department of Neurology, DMU Neurosciences, Paris, France. ✉e-mail: stanley.durrleman@inria.fr

It is particularly difficult to identify the most appropriate time frame for a disease like AD, which progresses over decades, in a non-linear manner, and with different clinical presentations between patients. The thresholds currently used for the main biomarkers and clinical endpoints are not sufficiently effective for the selection of patient populations with homogeneous progression profiles[5]. Disease modeling uses computational and statistical methods to address this question[6–14]. These models learn the variability of disease progression from observational longitudinal cohort data and can then predict the progression of patients from their historical data. They require various clinical or biomarker assessments at one or several time points as input. These techniques are beginning to be evaluated for clinical trial design. For example, a retrospective analysis showed that the effect size of treatment could be increased by targeting participants with a predicted type of progression at trial entry[15]. Other studies indicated that predicting the value of endpoints might make it possible to reduce sample sizes in clinical trials[16,17].

We propose here a software tool using a disease progression model for participant selection in clinical trials. The goal is to enrich the selected population of participants likely to display progression during the trial, a concept called prognostic enrichment[18] by the FDA and already applied in some AD trials[19,20]. We will use AD Course Map as a disease progression model. It is a non-linear mixed-effect model, which predicts both the dynamics of progression and the clinical presentation of the disease[21,22]. This technique outperformed the 56 alternative methods for predicting cognitive decline in the framework of the TADPOLE challenge[6,23]. We will compare this model with RNN-AD, which is a recurrent neural network, namely a deep learning method that learns temporal dynamic behavior. In June 2020, it ranked 2nd for the prediction of cognitive decline in the TADPOLE challenge[24].

We will first evaluate the ability of the model to predict progression for the main endpoints used as outcomes in current clinical trials. We will use five independent data sets with data from more than 4600 patients spread over four continents. We will analyze the systematic biases of such algorithms, their robustness to missing data, and suitability for generalization across countries, ethnicities, and disease stages. Finally, we will simulate inclusion procedures for clinical trials by varying several key parameters: the chosen outcome, trial duration, and selection criteria. Finally, we will show that participants predicted to be at risk of the outcome worsening constitute a population likely to show a greater and more homogeneous response to treatment.

## Results

### Characteristics of the study population
We used data from 4687 participants from five longitudinal multi-center cohorts from North America, Australia, Japan, and Europe: the Alzheimer's disease neuroimaging initiative (ADNI)[25–31] ($N = 1652$), the Australian imaging, biomarker and lifestyle flagship study of aging (AIBL)[32,33] ($N = 460$), the Japanese Alzheimer's disease neuroimaging initiative (J-ADNI)[34,35] ($N = 470$), the PharmaCog cohort[36,37] ($N = 111$) and the MEMENTO cohort[38] ($N = 1994$). Each study enrolled participants attending memory clinics.

Tables 1 and 2 summarize the characteristics of each data set. These data sets contain diverse patient profiles from different ethnic, genetic, and geographic backgrounds, with follow-up visits at different disease stages. For all these studies, the neuropsychological examinations were performed in accordance with international standards, and the image acquisition procedures were performed in accordance with the protocols established by the ADNI consortium. Together, these data sets, therefore, correspond to a relevant pool of patients for simulating inclusion procedures for a typical large multicenter phase III trial.

### Disease progression models learn the timing of changes in biomarker levels during disease progression
We train disease progression models using the ADNI participants with confirmed pathological amyloid levels as the training set ($N = 866$) with baseline and all available follow-up data. We kept the data from the other ADNI participants and the members of the four external cohorts as the validation set ($N = 3821$). The same protocol for training and validating the models is used for AD Course Map and RNN-AD. See Methods for details.

The two models include the following endpoints: Mini-Mental State Examination (MMSE), Alzheimer's Disease Assessment Scale–cognitive sub-scale with 13 items (ADAS-Cog13), Clinical Dementia Rating–sum of boxes (CDR-SB), volumes of the left and right hippo-campus and lateral ventricles, $A\beta_{1-42}$ and $p\text{-tau}_{181}$ levels in the cerebrospinal fluid (CSF), standard uptake value ratio (SUVR) for Amyloid PET and Tau PET scans. See Methods for details.

AD Course Map assumes that these endpoints follow a logistic progression curve during disease progression with distinct progression rate and age at the inflexion point[21,22]. It learns how this set of logistic curves need to be adjusted to fit individual data by changing the dynamic of progression and disease presentation (i.e., the relative value of the endpoints at a given disease stage). By contrast, RNN-AD learns how the values of the endpoints will change in the next month given the values of the endpoint at a given time-point. The 1-month transition is assumed to be a non-linear function (e.g. a neural network) of the current value of the endpoints and the current diagnosis. Supplementary Table 1 shows the goodness-of-fit on the training set, consistent with the results of our previous studies on AD Course Map[6] and RNN-AD[24]. See Methods for details.

### Disease progression models forecast cognitive decline
The disease progression models predict the subject-specific trajectory of biomarker changes from data collected from the subject concerned at one or several visits. The predicted trajectory is used to forecast the values of the biomarkers at future time points. Figure 1 illustrates this forecast procedure.

We repeatedly assessed the errors of AD Course Map and RNN-AD for forecasting cognitive endpoints (ADAS-Cog13, MMSE and CDR-SB) for participants in the validation set. We blinded the latest visits of the participants and tried to predict them from the unblinded data (see Supplementary Fig. 1 and Methods for details of the procedure). From 44,435 forecasts for ADAS-Cog13 (96,970 for MMSE and 96,849 for CDR-SB), we determined the absolute difference between predicted and actual results as a function of the characteristics of the participants and the information used for forecasting purposes.

Figure 2 shows the distribution of mean absolute errors (MAE) for AD Course Map and RNN-AD adjusted for co-founding factors. The reported errors are for the reference participant in the reference forecast design: a 75-year-old American woman from the ADNI cohort with an average education level, no APOE-ε4 mutations, and an A + T + N + status with a questionable dementia (CDR = 0.5 noted C-), for whom we forecast neuropsychological assessments in three years' time, based on two past visits separated by eight months with no missing data. AD Course Map yields a mean absolute error of 5.98 (95% CI = [5.44, 6.48]) on a scale of 85 for ADAS-Cog13, of 2.54 (95% CI = [2.39, 2.71]) on a scale of 30 for the MMSE, and of 1.86 (95% CI = [1.75, 1.99]) on a scale of 18 for the CDR-SB.

On all occasions, AD Course Map and RNN-AD yielded significantly smaller errors than two alternative methods: no-change prediction (predicting the same value as obtained at the participant's last visit) and a linear mixed-effects model ($p < 0.01$ for both, see Supplementary Table 2). These two alternatives were shown to be good predictor of short-term progression, essentially because of the overall slow pace of progression of the disease[6,23]. The deep learning method RNN-AD yields intermediate performance with adjusted mean

**Table 1 | Characteristics of study participants**

| | ADNI | AIBL | PHARMACOG | J-ADNI | MEMENTO |
|---|---|---|---|---|---|
| Number of subjects | 1652 | 460 | 111 | 470 | 1994 |
| Number of visits | 6.1 ± 3.0 [2, 17] | 3.7 ± 0.7 [2, 4] | 5.1 ± 0.6 [3, 7] | 5.1 ± 0.8 [4, 6] | 6.9 ± 1.8 [2, 9] |
| Follow-up duration (y) | 4.8 ± 3.1 [1.4, 15.2] | 4.1 ± 1.0 [1.5, 4.5] | 2.0 ± 0.3 [1.5, 3.0] | 2.7 ± 0.5 [1.5, 3.0] | 3.8 ± 0.7 [1.4, 5.2] |
| Time between visits (m) | 11.3 ± 6.6 [1.8, 62.8] | 18.2 ± 2.0 [18.0, 54.0] | 6.0 ± 0.6 [6.0, 18.0] | 7.8 ± 2.8 [6.0, 24.0] | 7.7 ± 3.4 [1.5, 53.7] |
| Age at baseline (y) | 73.3 ± 7.0 [55.1, 91.5] | 71.5 ± 7.1 [55.3, 92.1] | 69.8 ± 7.4 [50.5, 84.5] | 71.8 ± 6.7 [30.0, 85.0] | 70.6 ± 8.6 [32.5, 92.6] |
| Female | 771 (46.7 %) | 247 (53.7 %) | 63 (56.8 %) | 247 (52.6 %) | 1215 (60.9 %) |
| Education level | | | | | |
| ≤9 years | 24 (1.5 %) | 75 (16.3 %) | 47 (42.3 %) | 64 (13.6 %) | 355 (17.8 %) |
| Between 10 and 15 years | 525 (31.8 %) | 278 (60.4 %) | 39 (35.1 %) | 255 (54.3 %) | 1016 (51.0 %) |
| ≥16 years | 1103 (66.8 %) | 107 (23.3 %) | 25 (22.5 %) | 151 (32.1 %) | 571 (28.6 %) |
| Missing | | | | | 52 (2.6 %) |
| APOE-ε4 copies | | | | | |
| 0 | 917 (55.5 %) | 295 (64.1 %) | 56 (50.5 %) | 251 (53.4 %) | 1340 (67.2 %) |
| 1 | 588 (35.6 %) | 138 (30.0 %) | 41 (36.9 %) | 176 (37.4 %) | 500 (25.1 %) |
| 2 | 144 (8.7 %) | 27 (5.9 %) | 10 (9.0 %) | 40 (8.5 %) | 66 (3.3 %) |
| Missing | 3 (0.2 %) | | 4 (3.6 %) | 3 (0.6 %) | 88 (4.4 %) |
| Diagnosis at baseline | | | | | |
| CU | 649 (39.3 %) | 365 (79.3 %) | | 140 (29.8 %) | 831 (41.7 %) |
| MCI | 803 (48.6 %) | 59 (12.8 %) | 111 (100.0 %) | 211 (44.9 %) | 1163 (58.3 %) |
| Dementia | 200 (12.1 %) | 36 (7.8 %) | | 119 (25.3 %) | |
| A/T/N/C profile (worst for all visits) | | | | | |
| A-T-N-C- | 133 (8.1 %) | 91 (19.8 %) | | 24 (5.1 %) | 104 (5.2 %) |
| A*T*N*C- | 45 (2.7 %) | 122 (26.5 %) | | 82 (17.4 %) | 129 (6.5 %) |
| A+T-N-C- | 80 (4.8 %) | 48 (10.4 %) | | 6 (1.3 %) | 26 (1.3 %) |
| A+T+N-C- | 27 (1.6 %) | 4 (0.9 %) | | 1 (0.2 %) | 2 (0.1 %) |
| A+T+N+C- | 47 (2.8 %) | 11 (2.4 %) | | 1 (0.2 %) | 24 (1.2 %) |
| A*T*N*C~ | 80 (4.8 %) | 52 (11.3 %) | 2 (1.8 %) | 95 (20.2 %) | 880 (44.1 %) |
| A+T+N+C~ | 191 (11.6 %) | 6 (1.3 %) | 35 (31.5 %) | 28 (6.0 %) | 109 (5.5 %) |
| A*T*N*C+ | 111 (6.7 %) | 49 (10.7 %) | | 124 (26.4 %) | 118 (5.9 %) |
| A+T+N+C+ | 300 (18.2 %) | 10 (2.2 %) | 5 (4.5 %) | 45 (9.6 %) | 40 (2.0 %) |
| A+[T- or N-] C[~ or +] | 221 (13.4 %) | 13 (2.8 %) | 28 (25.2 %) | 15 (3.2 %) | 84 (4.2 %) |
| A-T+* | 183 (11.1 %) | 26 (5.7 %) | 8 (7.2 %) | 10 (2.1 %) | 56 (2.8 %) |
| A-T-N+C- | 40 (2.4 %) | 13 (2.8 %) | | 12 (2.6 %) | 81 (4.1 %) |
| A-T*N* C[~ or +] | 194 (11.7 %) | 15 (3.3 %) | 33 (29.7 %) | 27 (5.7 %) | 341 (17.1 %) |

Format for continuous variables: mean ± standard deviation [lowest, highest]. Stars in A(myloid)/T(au)/N(eurodegeneration)/C(linical) classification indicate unknown status (see Methods for details). (y) years, (m) months. *APOE* apolipoprotein E, *CU* cognitively unimpaired (CDR = 0 for MEMENTO), *MCI* mild cognitive impairment (CDR = 0.5 for MEMENTO), *CDR* clinical dementia rating.

absolute errors of 6.53 (95% CI = [6.02, 7.19]), 2.75 (95% CI = [2.57, 2.92]), and 1.95 (95% CI = [1.81, 2.09]) for the prediction of ADAS-Cog13, MMSE and CDR-SB respectively.

We investigated the change in MAE for ADAS-Cog 13 score for different categories of participants and forecast designs (Fig. 3). For AD Course Map, the number of previous visits considered (1, 2, or 3) did not significantly affect forecasting error. By contrast, for every additional year of time to prediction, MAE for ADAS-Cog13 score increased by 0.80 (95% CI = [0.71, 0.93]). Forecasts were not significantly affected by sex nor APOE genotype but were slightly improved for participants who are older than average and had longer education. On average, the forecasts for the European participants from PharmaCog cohort as well as the Japanese participants from the J-ADNI cohort were better than those for the American participants from the ADNI cohort, by about 1.1 and 0.6 points respectively. Forecasts were robust to missing CSF or Tau PET data, and slightly worsened when MRI or Amyloid PET were missing with differences in MAE of 0.27 (95% CI = [0.00, 0.55]) and 0.54 points (95% CI = [0.15, 1.02]) respectively. The model forecasts better at earlier stages of the AD continuum than at later clinical stages (Fig. 3a). The method was readily generalizable to the included participants with suspected non-

amyloid pathology (SNAP) and possible concomitant pathological non-Alzheimer's changes.

Similar conclusions were drawn for predictions of MMSE and CDR-SB (see Supplementary Figs. 2 and 3). AD Course Map performed better on all but one external validation cohort. Errors were robust to changes in the available information used to make the prediction, such as the number of unblinded visits and missing data. This method did not produce biased forecasts for women. Forecasts for those two endpoints however displayed slightly worse results for participants older than average or with an education level that is below the average, and for APOE-ε4 carriers.

**Disease progression models select participants displaying progression for trials**

We now use disease progression models to identify the participants likely to experience significant cognitive decline during a trial (see Fig. 4). The definition of participants displaying progression depends on the endpoint used to measure the condition and the duration of the trial. We simulated six clinical trials with different primary outcomes, trial durations, and inclusion criteria. These designs were inspired by real phase III trials (see Table 3).

**Table 2 | Distribution of endpoints for study participants**

| | ADNI | AIBL | PHARMACOG | J-ADNI | MEMENTO |
|---|---|---|---|---|---|
| CDR (global) | 0.4 ±0.5 [0.0, 3.0] (96.6 %) | 0.2 ±0.4 [0.0, 3.0] (99.6 %) | 0.5 ±0.1 [0.0, 1.0] (88.4 %) | 0.5 ±0.4 [0.0, 3.0] (99.9 %) | 0.3 ±0.3 [0.0, 3.0] (96.9 %) |
| CDR (sum of boxes) | 2.0 ±2.8 [0.0, 18.0] (96.6 %) | 0.9 ±2.5 [0.0, 18.0] (99.6 %) | / | 2.5 ±2.6 [0.0, 18.0] (99.8 %) | 0.8 ±1.6 [0.0, 18.0] (96.4 %) |
| MMSE | 26.9 ±3.9 [0.0, 30.0] (94.9 %) | 27.5 ±4.1 [0.0, 30.0] (99.8 %) | 26.2 ±2.7 [10.0, 30.0] | 25.2 ±4.1 [1.0, 30.0] (99.8 %) | 27.8 ±2.7 [2.0, 30.0] (97.9 %) |
| ADAS-Cog13 | 16.4 ±11.5 [0.0, 85.0] (94.3 %) | / | 20.4 ±7.7 [2.7, 50.3] (97.1 %) | 19.8 ±11.3 [0.0, 68.0] (99.5 %) | / |
| Lateral ventricles volume (% ICV) | 2.63 ±1.26 [0.57, 8.64] (42.6 %) | 2.48 ±1.30 [0.59, 7.44] (47.3 %) | 2.58 ±1.12 [0.74, 6.75] (97.7 %) | 2.65 ±1.19 [0.54, 9.79] (93.3 %) | 2.25 ±1.16 [0.36, 9.32] (25.2 %) |
| Hippocampus volume (% ICV) | 0.47 ±0.07 [0.23, 0.72] (42.6 %) | 0.47 ±0.06 [0.26, 0.69] (47.4 %) | 0.46 ±0.10 [0.17, 0.67] (97.7 %) | 0.42 ±0.09 [0.16, 0.65] (93.3 %) | 0.48 ±0.08 [0.19, 0.87] (25.2 %) |
| $Abeta_{1\text{-}42}$ level in CSF ($) | −0.35 ±1.16 [−1.97, 6.56] (25.6 %) | 0.61 ±1.15 [−1.51, 3.56] (7.1 %) | −0.53 ±1.17 [−2.68, 2.53] (19.4 %) | −0.23 ±1.25 [−2.57, 3.09] (12.7 %) | −0.14 ±1.10 [−2.11, 3.16] (3.6 %) |
| $p\text{-}Tau_{181}$ level in CSF ($) | 0.22 ±1.03 [−1.19, 7.07] (25.6 %) | −0.34 ±0.81 [−1.96, 2.15] (7.1 %) | 0.07 ±1.02 [−1.13, 4.59] (19.4 %) | 0.13 ±1.16 [−1.62, 4.04] (12.7 %) | 0.02 ±1.04 [−1.66, 6.74] (3.6 %) |
| Total Tau level in CSF ($) | 0.23 ±1.02 [−1.44, 6.19] (25.6 %) | −0.89 ±0.93 [−1.89, 6.77] (7.1 %) | 0.07 ±0.91 [−1.02, 4.44] (19.4 %) | / | −0.03 ±1.07 [−1.17, 5.97] (3.6 %) |
| Amyloid PET (CL) | 36.6 ±44.4 [−33.6, 213.2] (28.2 %) | / (*) | / | / | / (*) |
| Tau PET (SUVR) | 1.58 ±0.31 [1.14, 4.64] (9.0 %) | / | / | / | / |

Data are reported as mean value ± standard deviation [lowest, highest] (% available data [% available data when some data is missing]), or "/" if variable is not available at all. CDR Clinical Dementia Rating, MMSE Mini-Mental State Examination, ADAS-Cog13 Alzheimer's Disease Assessment Scale-cognitive sub-scale (13 items), ICV intracranial volume, CSF cerebrospinal fluid, ($) in harmonized units (see Methods), PET positron emission tomography, CL centiloid scale, (*) Amyloid PET data for the AIBL and MEMENTO cohorts were used only for the determination of the amyloid status, SUVR standardized uptake value ratio.

For each trial, we selected the participants in the validation set who met the inclusion criteria at one of their visits (considered as the baseline visit for the simulated trial) and attended a follow-up visit after a period equal to the theoretical duration of the trial. We split this population into two equal halves: fast and slow progressors, according to whether the outcome considered (e.g. the annual change in endpoint relative to baseline) was above or below the population median value. We aimed to identify the participants in these two groups exclusively on the basis of their baseline data.

We used the disease progression models to forecast the values of the endpoint at the end of the trial from the baseline data for each participant. The predicted outcome was used as a prognostic score. For AD Course Map, Pearson correlations with the true outcome range from 28% to 47% depending on the trial, while for RNN-AD they range from 13% to 36% (see Supplementary Table 3). Participants with a prognostic score above a given threshold were considered to be likely to be fast progressors. We plotted receiver operating characteristics (ROC) curves for the six simulated trials (Fig. 5). The area under the ROC curve (AUC) of the six simulated trials fell within the 65–80% range for AD Course Map and within the 55–80% range for RNN-AD (see Fig. 5).

We compared this prognostic enrichment strategy with two alternative methods: selecting participants at random (bisector of the ROC curve) as currently done in most trials, or selecting participants based on their APOE genotype (gray crosses in Fig. 5). All selection methods were significantly better than random selection, meaning that disease progression models succeed in identifying the progressors compared to the current practice that does make any difference among the participants meeting the inclusion criteria. In all but one case, selections with AD Course Map were significantly better than selection on the basis of APOE genotype. RNN-AD also compares favorably against the two alternatives. Nevertheless, it has significantly worse performance than AD Course Map in two out of six tested scenarios, with a drop of 9% and 14% in the ROC AUC. AD Course Map shows therefore more robust results than RNN-AD when the trial design is varied.

We analyzed whether our assessment of the risk of progression led to an over- or under-selection of certain types of participants relative to the true progressors (see Supplementary Fig. 4). Depending on the design, the group that was selected using AD Course Map displayed slight enrichment in men or women, and tended to be biased towards older participants. The selected participants were often, but not always, enriched in carriers of the APOE-ε4 variant. The presented disease progression models do not use sociodemographic or genetic factors as proxies for the selection of participants displaying progression. They limit therefore the biases of sex, age, or APOE-ε4 carriership, which are the basis of current practices to increase the likelihood that a participant progresses during a trial.

### Disease progression models can be used to design more powered clinical trials

The automatic selection of participants displaying progression makes it possible to implement prognostic enrichment strategies in trials (see Fig. 4). For each trial design, we simulated a hypothetical treatment decreasing the outcome value. We calculated the sample size required to show the effect of this treatment for a range of treatment effects (see Methods). We compared the results when all eligible participants were included to those obtained when only participants predicted to be fast progressors at baseline were included.

We plotted sample size against the treatment effect for all six simulated trials (Fig. 6). The selection of participants at risk of progression with AD Course Map allowed a significant reduction in sample size relative to current inclusion criteria alone, across all scenarios tested. For a treatment effect of 25%, the sample size was reduced by

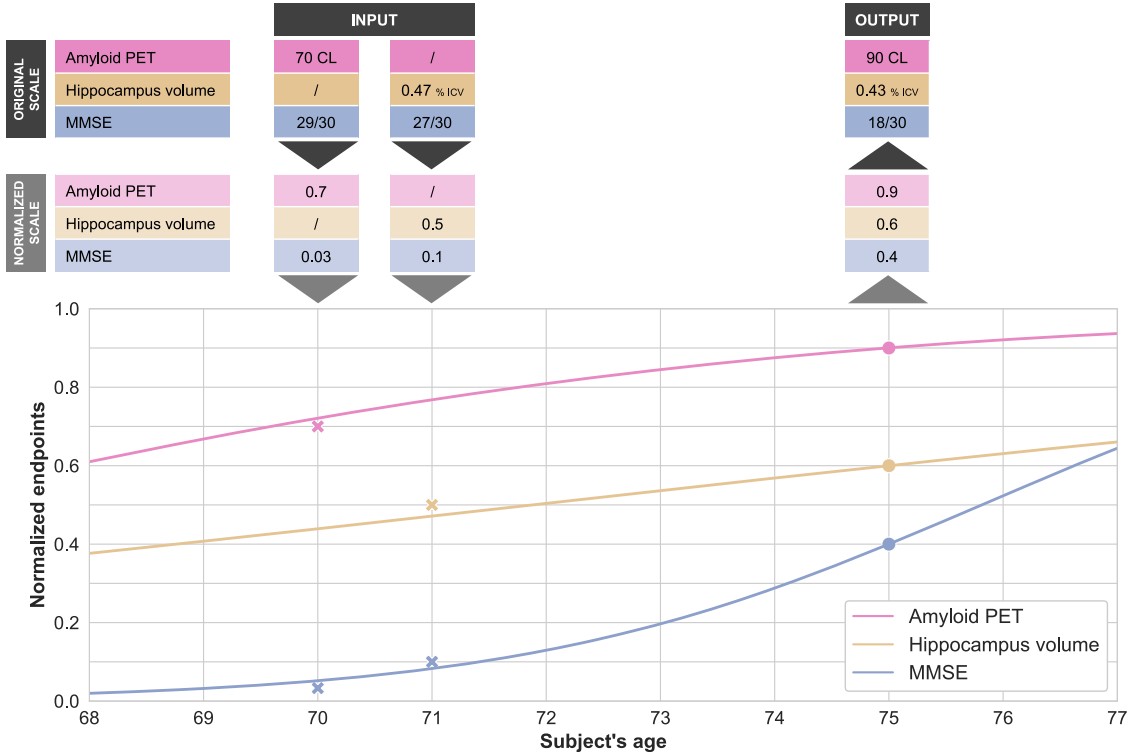

**Fig. 1 | Disease progression models forecast the progression of endpoints from historical data of a participant.** In this simplified example, the model has only three endpoints (Amyloid PET, Hippocampus volume, and mini-mental state examination (MMSE)). The participant has been observed twice at 70 and 71 years old (colored crosses). After normalizing the data to a 0–1 scale (0 being the most normal and 1 the maximum pathological change), the model predicts the participant-specific progression curves. From these curves, one forecasts the values of the three endpoints in 4 years' time (colored dots). As shown in this example, AD Course Map does not require the imputation of missing data. In trial simulations, the curves are predicted from the data at a single time point, e.g. the baseline. CL centiloid scale, ICV intracranial volume.

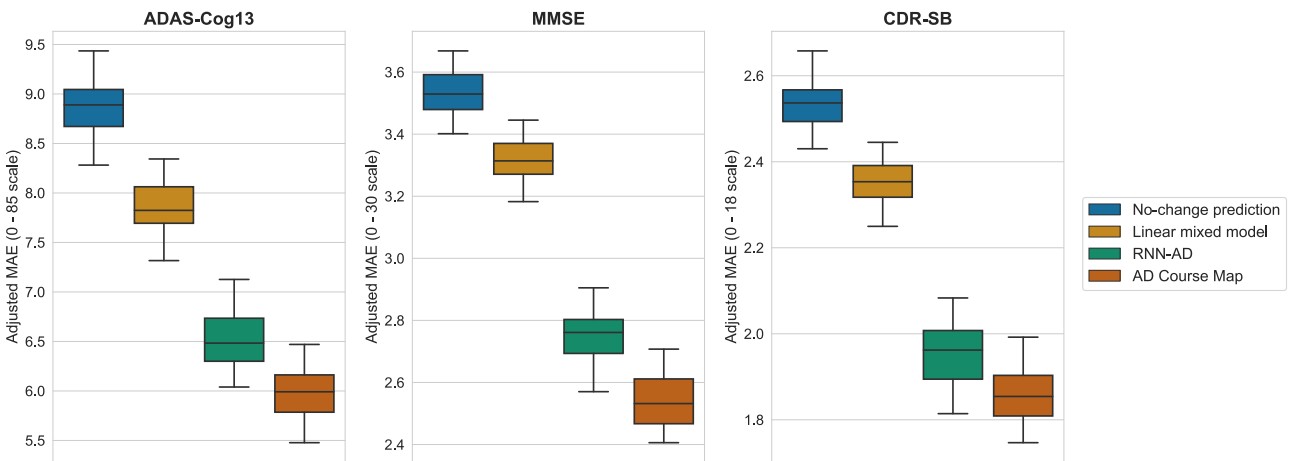

**Fig. 2 | AD Course Map forecasts cognitive decline better than alternative methods.** The mean absolute error is reported for the reference participant: a 75-year-old American woman from ADNI with an average level of education, no APOE-ε4 mutation, and a A + T + N + C- status (i.e., with CDR global of 0.5), for whom we forecast neuropsychological assessments in three years' time, based on two past visits separated by eight months and for which all data were available. Box plots represent median value, first and third quartiles; whiskers represent the empirical 95% confidence interval. Statistics are computed for $n = 100$ resampling of the validation set (see Methods). Source data are provided as a Source Data file. MAE mean absolute error.

50.2% (±7.1) for participants at risk of the onset of AD, by 40.9% (±4.9) for a trial targeting individuals with preclinical AD and high brain amyloid levels, by between 38.1% (±1.6) and 45.4% (±2.0), depending on the outcome considered, for subjects with early AD and high levels of brain amyloid, by 44.6% (±3.9) for subjects with early AD and high brain levels of tau, and by 43.1% (±0.8) for participants with mild cognitive impairment probably due to AD or mild AD.

For all preclinical and early AD trials, enrichments based on AD Course Map significantly outperformed the selection of APOE-ε4 variant carriers only. For mild cognitive impairment due to AD or a mild AD trial, the performance of enrichment based on AD Course Map was not significantly different from targeting APOE-ε4 carriers. AD Course Map achieved a similar decrease in sample size, but without the need to target a specific genetic profile. In this case, we also found that 49.2%

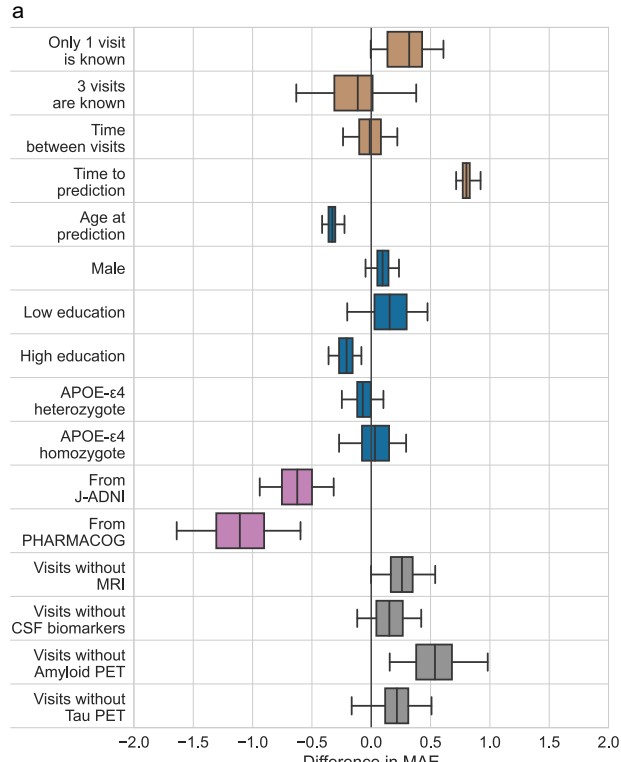

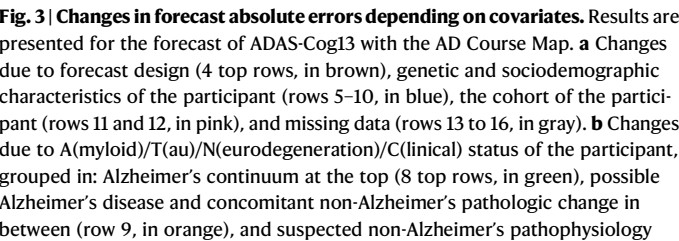

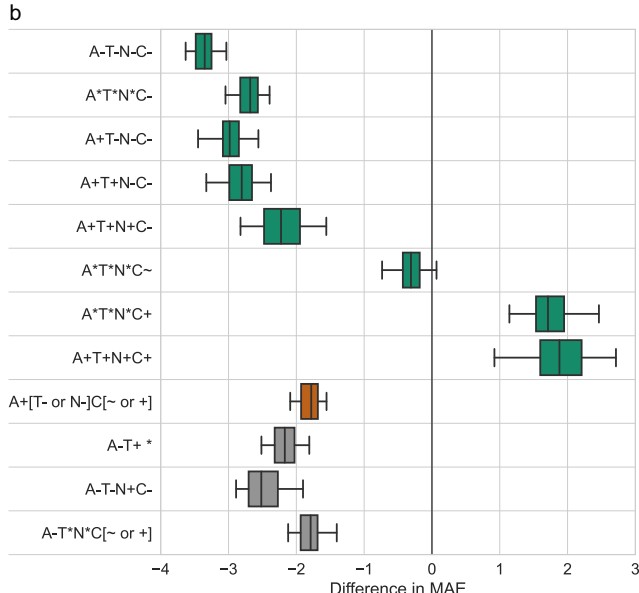

**Fig. 3 | Changes in forecast absolute errors depending on covariates.** Results are presented for the forecast of ADAS-Cog13 with the AD Course Map. **a** Changes due to forecast design (4 top rows, in brown), genetic and sociodemographic characteristics of the participant (rows 5–10, in blue), the cohort of the participant (rows 11 and 12, in pink), and missing data (rows 13 to 16, in gray). **b** Changes due to A(myloid)/T(au)/N(eurodegeneration)/C(linical) status of the participant, grouped in: Alzheimer's continuum at the top (8 top rows, in green), possible Alzheimer's disease and concomitant non-Alzheimer's pathologic change in between (row 9, in orange), and suspected non-Alzheimer's pathophysiology (SNAP) at the bottom (3 bottom rows, in gray). Coefficients below zero indicate a lower mean absolute error (MAE) (better forecast) than those for the reference participant and design. For example, if the reference participant comes from J-ADNI instead of ADNI, the prediction of ADAS-Cog13 is more accurate, resulting in a 0.63 point decrease in MAE (95% CI = [0.32, 0.96]). Box plots represent median value, first and third quartiles; whiskers represent the empirical 95% confidence interval. Statistics are computed for $n = 100$ resampling of the validation set (see Methods). Source data are provided as a Source Data file. MAE mean absolute error.

(95% CI = [48.6, 49.9]) of the participants would be selected by AD Course Map, versus 39.1% (95% CI = [38.7, 39.4]) for heterozygous APOE-ε4 carriers, facilitating recruitment with AD Course Map (see Supplementary Table 4).

RNN-AD also allowed a significant reduction of the sample size compared to current practice, from 21% to 42% depending on the tested scenario. Nevertheless, the reduction was never better than with AD Course Map with an increase of 10% and 35% participants to be selected for the two scenarios where RNN-AD yielded a lower AUC (see Supplementary Table 5).

## Discussion

We used disease progression models to forecast cognitive decline across all stages of the AD continuum. Using five independent cohorts containing more than 4,600 participants, we show here that AD Course Map provides a fair, robust, and generalizable predictive method. It is fair, in that its predictions are not biased with respect to sex, and are only marginally affected by level of education and the age of the participant. The method is robust to missing CSF or Tau PET biomarkers, but in general better results are achieved when MRI and Amyloid PET data are present. The model was trained on data acquired in North America, but it is readily generalizable to participants from Europe, Asia, and Oceania, with no loss of performance. It performed better at the earliest preclinical stages of the AD continuum than at later disease stages, and is therefore relevant for early-stage interventions.

Disease progression models automatically identify the participants already at risk of experiencing cognitive decline at baseline in a trial. They can therefore be used to enrich the trial population in participants likely to experience a worsening of a given endpoint during the trial. By targeting more homogeneous groups of participants displaying progression, AD Course Map makes it possible to decrease sample size significantly, by 38% up to 50%, at the expense of discarding about half of the screened participants. It shows better and more robust performance than the deep learning method RNN-AD. Disease progression models adapt seamlessly to various clinical trial designs targeting different disease stages with different outcomes and trial durations. They do so without the need to re-train the model for each new trial. In comparison, a recent method based on another prognosis score reported sample size reductions of 20% to 28%[17].

The main limitation of the method is the data used to monitor disease progression. Cognitive assessment displays about 10% inter-rater variability[39–41]. MRI biomarkers also display a similar degree of variability between two scans acquired on the same day for the same participant, and their reliability is further decreased by possible variations in the processing pipelines[42]. Mapping CSF biomarkers from different immunoassays also limit their reliability[43]. These factors limit the accuracy of the method for forecasting disease progression. Increasing the reliability of these measurements would improve the performance of the approach described here. In the future, disease progression models such as AD Course Map may also benefit from the inclusion of promising new biomarkers, such as plasma biomarkers, neurofilament light chain[44], or digital biomarkers[45].

Given these limitations, it is notable that such large sample size reductions can be achieved with data already available in routine

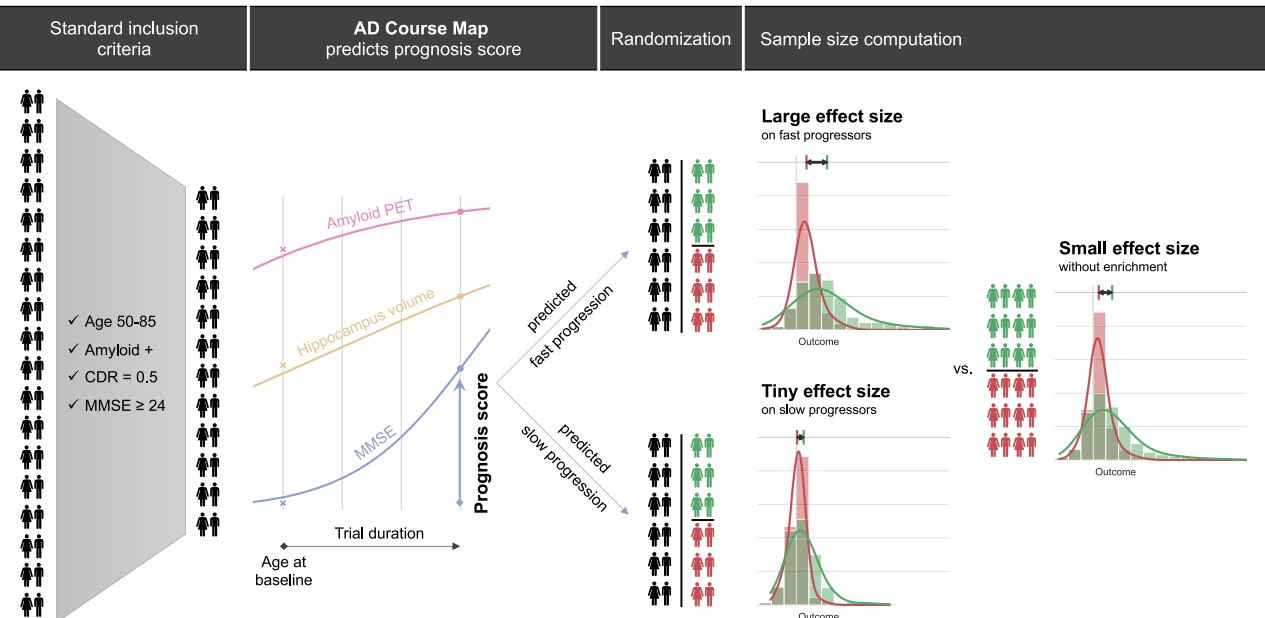

**Fig. 4 | Illustration of the prognostic enrichment procedure in a clinical trial.** Participants are selected first using standard inclusion criteria and undergo a series of exams. A disease progression model, such as AD Course Map, then forecasts the progression of each participant's data and predicts if the participant is likely to progress significantly during the trial, as measured by the predicted outcome change, which is the mini-mental state examination (MMSE) in this example. The treatment effect (e.g., a 25% reduction of the change of the MMSE during trial) leads to a greater effect size, and therefore a smaller sample size, on the group of predicted fast progressors compared to the group of predicted slow progressors or the two groups combined. As a result, one may demonstrate the treatment efficacy with fewer participants by monitoring only the group of predicted fast progressors.

## Table 3 | Description of the simulated trials

| Clinical trial description | Inclusion/exclusion criteria | Primary outcome (annual rate of change of...) | Trial duration window | Inspiration from existing AD trial (Clinical-Trials.gov identifier) |
|---|---|---|---|---|
| Participants at risk of AD onset | - Age [59.9, 76.1]<br>- CDR global = 0<br>- MMSE ≥ 24<br>− 1 risk factor of:<br>> Homozygous APOE-ε4<br>> Heterozygous APOE-ε4 & Amyloid+ (*) | MMSE | 4 years<br>± 12 months | Novartis<br>Generation S2 (NCT03131453) |
| Preclinical AD with high brain amyloid levels | - Age [54.9, 81.1]<br>- CDR global = 0<br>- MMSE ≥ 27<br>- Amyloid+ (*) | ADAS-Cog13 | 4 years<br>± 12 months | Eisai<br>AHEAD A45 (NCT04468659) |
| Early AD with high brain amyloid levels | - Age [49.9, 86.1]<br>- CDR global = 0.5<br>- MMSE ≥ 24<br>- Amyloid+ (*) | - MMSE<br>- CDR-SB | 1.5 years<br>± 6 months | Biogen<br>EMERGE / ENGAGE (NCT02477800 & NCT02484547) |
| Early AD with high brain tau levels | - Age [54.9, 81.1]<br>- CDR global = 0.5<br>- p-Tau+ (*) | ADAS-Cog13 | 4.5 years<br>± 6 months | Janssen<br>Autonomy<br>(NCT04619420) |
| MCI probably due to AD or mild AD | - Age [54.9, 86.1]<br>- CDR global = 0.5 or 1<br>- From AD data sets | MMSE | 3 years<br>± 9 months | / |

Six trials were simulated because we considered two possible primary outcomes for the trial targeting early Alzheimer's disease (AD) with high brain amyloid levels (third row). *MCI* mild cognitive impairment, *AD* Alzheimer's disease.
*CSF or PET (worst visit to date).

clinical practice. These findings demonstrate the benefits of companion software tools for patient recruitment in trials and for supporting clinicians in the future, enabling them to prescribe the right treatment to the right patient at the right time.

## Methods
### Participants
We used the data from five longitudinal multicenter cohorts: the ADNI[25–31] ($N = 1652$), the Australian imaging, biomarker, and lifestyle flagship study of aging (AIBL)[32,33] ($N = 460$), the JJ-ADNI[34,35] ($N = 470$), the PharmaCog cohort[36,37] ($N = 111$) and the MEMENTO cohort[38] ($N = 1994$).

The study protocols were approved by the ethical committees of the university of southern California (ADNI), Austin Health, St Vincent's Health, Hollywook Private Hospital and Edith Cowan University (AIBL), IRCCS Istituto Centro San Giovanni di Dio Fatebenefratelli (Pharma-Cog), Comité de protection des personnes sud-ouest et outre-mer III (MEMENTO), the National Bioscience Database Center Human Database (J-ADNI). Informed consent forms were obtained from research

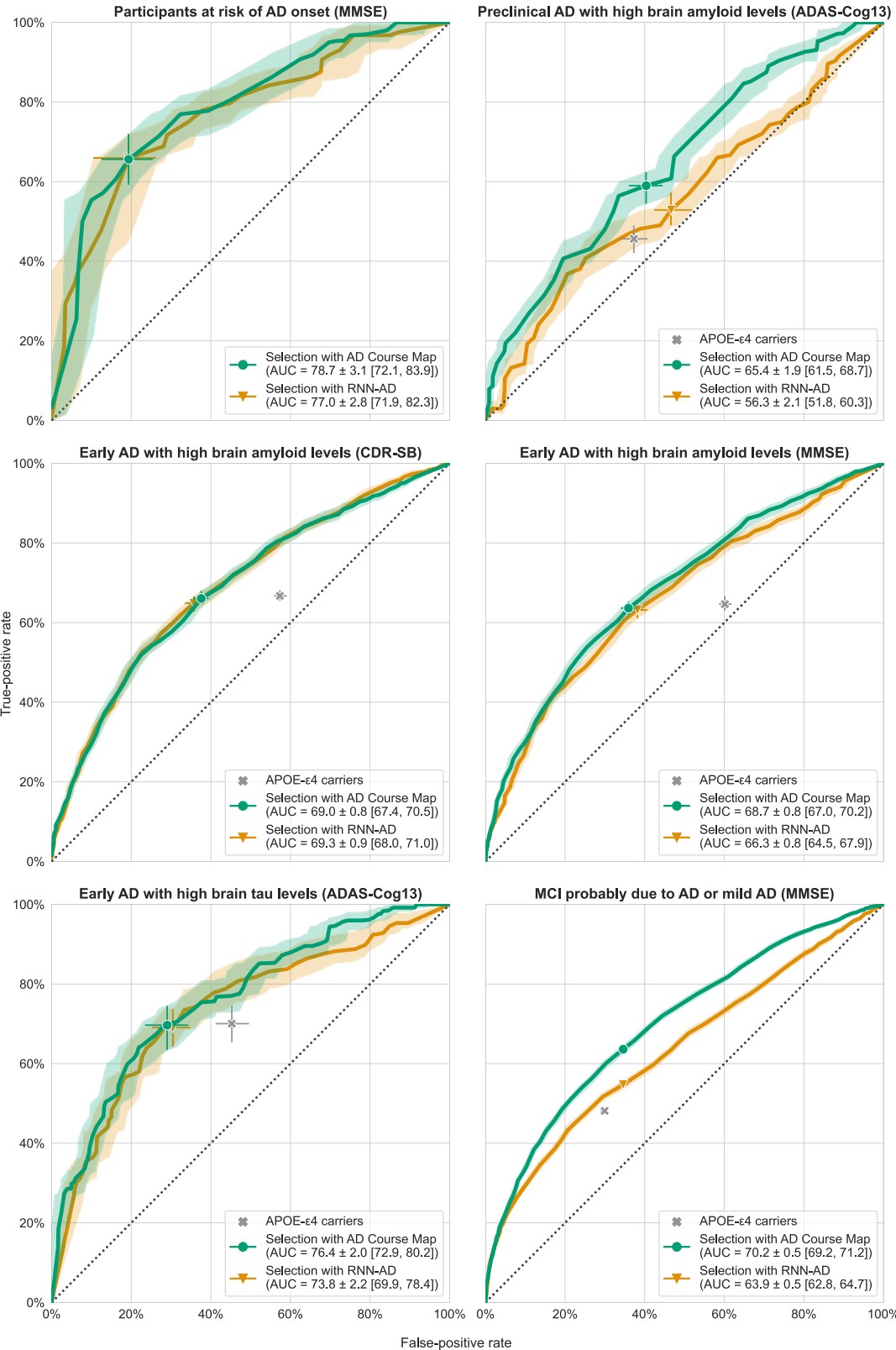

**Fig. 5 | AD Course Map and RNN-AD select participants at risk of experiencing a worsening of the outcome during the trial.** Receiver operating characteristic (ROC) curves are shown. They demonstrate the performance of AD Course Map and RNN-AD in selecting the group of participants with the largest change in primary outcome during follow-up. Shaded areas correspond to the empirical 95% confidence interval. The green circle and orange triangle on each curve correspond to selections splitting the participants into two equal groups, with bars representing the 95% confidence intervals. The cross in gray gives the specificity and sensitivity when APOE-ε4 carriers (with 1 or 2 copies) are selected, with bars indicating the 95% confidence interval (note: the first trial includes only APOE-ε4 carriers, and there is, therefore, no gray cross). Statistics are computed for $n = 100$ resampling of the validation set (see Methods). Source data are provided as a Source Data file. AUC: area under the ROC curve (mean ± standard deviation with 95% confidence interval).

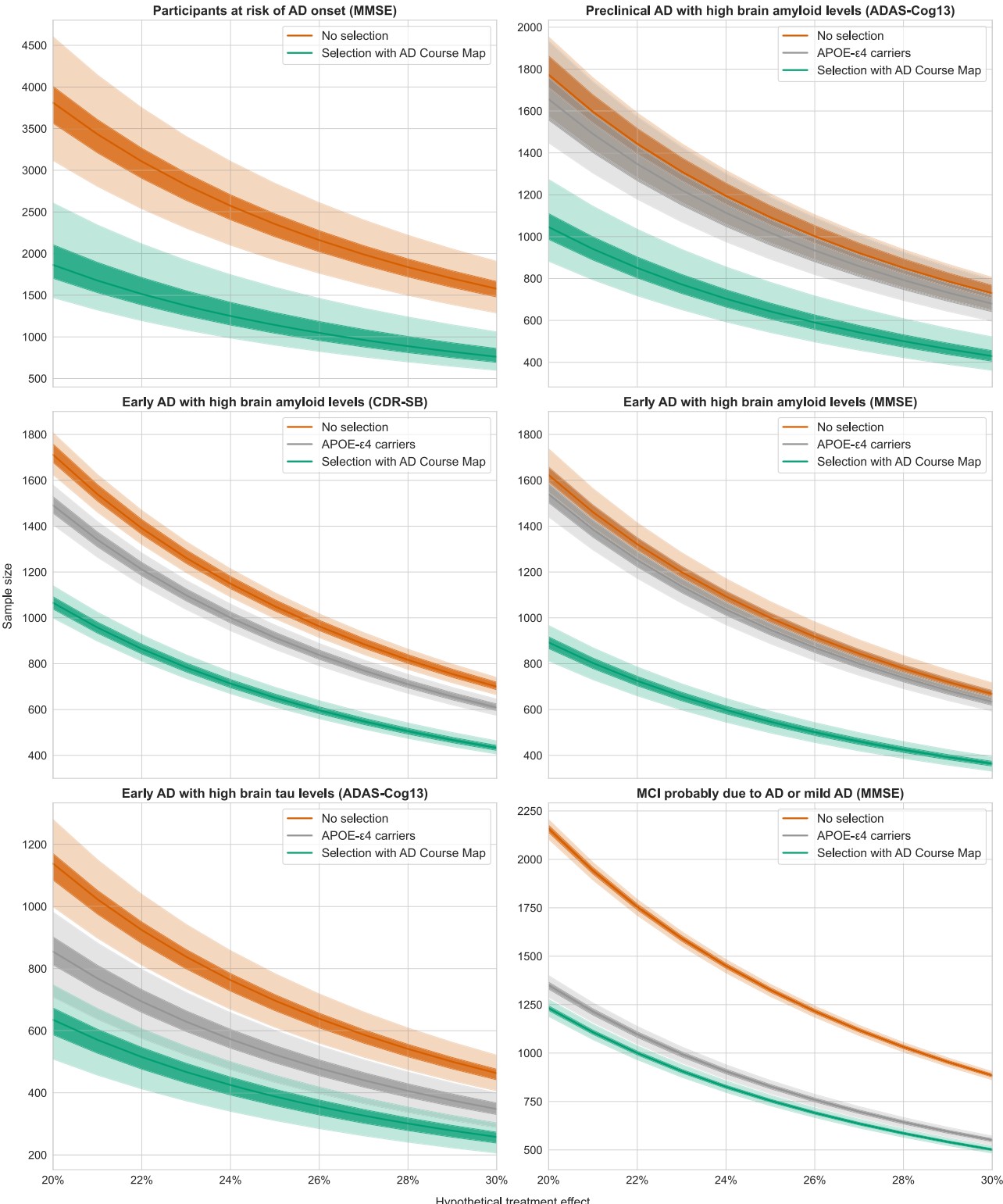

**Fig. 6 | Enrichment based on AD Course Map significantly decreases the sample size for a hypothetical treatment effect ranging from 20% to 30%.** Reported sample sizes are the total size for two arms. The light-shaded areas represent the 95% confidence interval and the dark-shaded areas the 50% confidence interval around the median value. For all preclinical and early Alzheimer's disease (AD) trials, enrichment based on AD Course Map significantly outperformed the enrichment based on APOE-ε4 carriership. Statistics are computed for $n = 100$ resampling of the validation set (see Methods). Source data are provided as a Source Data file.

participants. The research has been performed in accordance with the Declaration of Helsinki and relevant guidelines and regulations. Participants were not compensated for the current study.

The five cohorts are longitudinal observational studies with an average observation period ranging from 2.0 years for PHARMACOG to 4.8 years for ADNI, with an average number of visits ranging from 3.7 in AIBL to 6.9 in MEMENTO. We considered all participants with at least one year of follow-up. The sociodemographic, genetic, biological and clinical characteristics of the selected participants are reported in Tables 1 and 2, as well as the proportion of available data in each cohort.

## Neuropsychological assessments

In our experiments, we considered the following neuropsychological assessments:

- The mini-mental state examination[39] (MMSE),
- The Alzheimer's disease assessment scale–cognitive sub-scale with 13 items[40,46] (ADAS-Cog13),
- The clinical dementia rating scale[41,47] – sum of the boxes score (CDR-SB).

## Structural magnetic resonance imaging/anatomical imaging biomarkers

We extracted cortical and subcortical volumes from three-dimensional T1-weighted magnetization-prepared rapid gradient-echo imaging (MPRAGE) sequences.

For the ADNI study, scans were acquired in the standardized protocol for morphometric analyses (http://adni.loni.usc.edu/methods/documents/mri-protocols/). The ADNI MRI core processed raw scans, using Gradwarp for the correction of geometric distortion due to gradient nonlinearity[48], B1-correction for the adjustment of image intensity inhomogeneity[26], N3 bias field correction for reducing residual intensity inhomogeneity[49,50], and geometric scaling for adjusting scanner- and session-specific calibration errors[26,51]. The same MRI protocol was also used in AIBL[32], J-ADNI[52], PharmaCog[36], and MEMENTO[38,53].

For all studies, cortical reconstruction and volumetric segmentation were performed with the Freesurfer image analysis suite (http://surfer.nmr.mgh.harvard.edu/). Version 5.3 was used for J-ADNI, MEMENTO, and PharmaCog, and version 6.0 for ADNI and AIBL, operated within Clinica for reproducibility purposes[54]. The cohort effect in the following analyses accounts for possible differences due to different versions of the software.

We calculated the mean volume of the left and right hippocampus, and the total volume of the lateral ventricles (including inferior lateral volume). Hippocampus segmentation with Freesurfer was previously reported to have good reproducibility[55,56]. Both volumes were normalized by estimated total intracranial volume (ICV).

## Cerebrospinal fluid biomarkers

We used the concentrations in cerebrospinal fluid (CSF) of β-Amyloid 1–42 peptide ($A\beta_{1-42}$), Tau protein, phosphorylated at the threonine 181 residue (p-$Tau_{181}$), and total tau protein (t–Tau).

ADNI used the automated Elecsys immunoassay (Roche); AIBL, PharmaCog, and MEMENTO used INNOTEST single-analyte ELISA tests (Innogenetics/Fujirebio NV), and J-ADNI used the multiplex xMAP Luminex platform with the INNO-BIA AlzBio3 immunoassay kit (Innogenetics/Fujirebio NV).

We harmonized the measurements to account for the differences in immunoassays and participants' characteristics across cohorts. Within each cohort, we regressed each biomarker against age, APOE genotype, and CDR global score with a linear mixed model with random intercept. We then linearly transformed the measurements so that the intercept is 0 and the total variance is 1 for all cohorts. Harmonization equations used are listed in Supplementary Table 6 for reproducibility purposes.

## Positron emission tomography/functional imaging biomarkers

For ADNI participants, we used regional standardized uptake value ratios (SUVR) extracted from Amyloid PET scans ([18F]-Florbetapir and [18F]-Florbetaben radiotracers), and, starting from ADNI 3, Tau PET scans ([18F]-AV-1451 radiotracer). Each PET scan was registered together with the MRI for the subject performed as close as possible to the PET scan in terms of time.

For Amyloid PET, we used a cortical-summary region consisting of the frontal, anterior/posterior cingulate, lateral parietal, and lateral temporal regions; data were normalized with a composite reference region consisting of the whole cerebellum, brainstem/pons, and eroded subcortical white matter[57–59]. These PET SUVR values were converted to the centiloid scale (CL)[60] using equations from the literature[61] listed in Supplementary Table 6. In the AIBL cohort, the processed Amyloid PET SUVR data that correspond to the published centiloid conversion equations were not publicly available. In the MEMENTO cohort, Amyloid PET SUVR data are not directly comparable with ADNI data and equations for centiloid conversion were not available. Therefore, we used Amyloid PET data on these cohorts only to define the Amyloid status of the participants, using pathological thresholds provided by these studies.

For Tau PET, we used a volume-weighted average SUVR value for all anatomical Braak regions of interest (I-VI)[62], normalized against the inferior cerebellum gray matter[63].

## A/T/N/C classification

We classified participants with the A(myloid)/T(au)/N(eurodegeneration) classification[64,65], together with a C(ognition)/C(linical) group based on the Clinical Dementia Rating (CDR) global score (see the Supplementary Table 7 for all thresholds used). Participant category at a given visit was based on the patient's all-time worst biomarker levels to date. Incomplete A/T/N/C profiles are denoted with a star after any of the biomarkers that could not be determined.

## Disease progression models

We trained and tested two disease progression models: AD Course Map and RNN-AD. AD Course Map is built on the principles of a parametric Bayesian non-linear mixed-effects model[21,22]. RNN-AD is built on the principles of recurrent artificial neural networks[24,66]. The implementation of both models relies on the open-source software that was made publicly available by their respective authors.

Both models use the same set of endpoints as input: MMSE, CDR-SB, ADAS-Cog13, volume of the left and right hippocampus and lateral ventricles, CSF $A\beta_{1-42}$ and p-$tau_{181}$ levels, together with cortical-summary SUVR on Amyloid PET and Tau PET scans. They consider these endpoints at one or several visits of a participant, allowing for possible missing data, and predict the value of all these endpoints at any time-point in the future. AD Course Map also takes into account the age of the participant at each visit, while RNN-AD takes into account only the duration between two consecutive visits, irrespective of the age of the participant. In addition, RNN-AD needs the diagnosis of the participant at the corresponding visit, the diagnosis being cognitively normal, mild cognitive impairment, or demented, as defined in the ADNI protocol.

AD Course Map assumes that these endpoints follow a logistic progression curve during disease progression with distinct progression rate and age at the inflexion point[21,22]. It learns how this set of logistic curves needs to be adjusted to fit individual data, by changing the dynamic of progression and disease presentation (i.e., the ordering and timing of progression among the endpoints). The shape and position of the reference set of logistic curves are the fixed effects, and the parameters changing these curves to fit individual data are the random effects. The model parameters (fixed effects together with the mean and variance of the random effects) are estimated using a training data set containing the repeated measurements of a multitude of participants. After the training phase, the model is fit to the measurements of one test participant (outside the training test) at one or several visit, using the learnt distribution of the random effects as a regularizer. As a result, the model predicts a subject-specific set of logistic curves, which shows the value of each endpoint at any age of the participant.

By contrast, RNN-AD does not make any assumption on the life-long pattern of progression of the endpoints. It learns instead how the values of the endpoints will change in the next month given the values of the endpoint at a given time-point. The 1-month transition is

assumed to be a non-linear function of the current value of the endpoints and the current diagnosis (e.g. artificial neurons). The parameters of this transition function are estimated using a training data set containing the repeated measurements of a multitude of participants. After the training phase, the measurements of one test participant (outside the training set) at one or several visits are used as input of the model. The model then computes the values of all the endpoints at each month in the future.

AD Course Map can be trained and tested with missing data: the likelihood is optimized using the available data only. Model training is robust to missing data[6], so we did not perform data imputation. By contrast, RNN-AD needs complete data at the baseline visit. We imputed missing data with the mean value of the endpoint in the training set, following authors' recommendations;[24] missing data at subsequent visits are imputed recurrently using model predictions.

Both models also need an internal step of data normalization. For AD Course Map, cognitive assessments were normalized to a 0 to +1 scale according to the theoretical minimum and maximum values of each assessment, 0 representing the theoretical best value (unaffected participants) and +1 the worst possible value. Harmonized amyloid PET data are clipped between 0 and 100 and converted to a (0,1) scale. MRI, tau PET, and Harmonized CSF data were clipped at the first and last centile, and then linearly mapped to a (0,1) scale. For RNN-AD, normalization consists in a z-score transformation estimated from training data.

Regardless of the normalization procedure, the outputs of the models are always converted back to the native scale (and unit) of the measurement before being analyzed (see Fig. 1). Predicted values are therefore comparable with the true, non-normalized data. Forecast errors can be compared across methods that do not use the same normalization procedure.

## Validation procedure

We split the data sets in two (see Supplementary Fig. 1). We first considered the ADNI participants who were amyloid-positive according to CSF or PET data on at least one visit (shown in red in Supplementary Fig. 1). We then kept the other ADNI participants and all participants from the four other cohorts as an external validation set (shown in blue in the Supplementary Fig. 1).

We then split the amyloid-positive ADNI participants into five random folds and trained AD Course Map and RNN-AD using all available data of the participants in four out of the five folds, e.g., the training set. We repeated this procedure with another split, so that we ended up with 10 instances of each model. Each participant has been counted twice as a test subject in the left-out fold. Therefore, it can be used twice for evaluating prediction tasks with two different instances of each model. By contrast, each participant in the external validation set can be tested with 10 different instances of each model. In the following, we averaged the prediction made by the 2 instances of the model for the participants in the test sets, and by the 10 instances for the participants in the external validation set.

The test subjects did not contribute to any model selection or hyperparameter tuning neither for AD Course Map nor for RNN-AD. Therefore, we pooled the forecasts of test subjects with the ones in the external validation set.

## Forecasting endpoints

We aimed to assess the accuracy of each model to forecast the values of the endpoints of a participant in the test set or the external validation set. The general principle is to blind the latest data of the participant, use the unblinded data as input of the model, and compare the predicted value with the blinded data.

We used a combinatorial procedure to generate prediction tasks, as described in Supplementary Fig. 1. Because we have multiple follow-up visits, we assessed several forecast errors for a single participant: we

blinded the data of the participant except at one to three consecutive visits, we predict the individual trajectory using the unblinded data, and forecast the data at the blinded visits after the latest unblinded visit. We required that the participants are between 50 and 90 years old and have a CDR global of at most 2 at the latest unblinded visit to exclude severely demented participants, and that the blinded visits used to assess the forecast fall between 1.4 and 6.6 years after the latest unblinded visit. We computed the forecast error as the absolute difference between this value and the value of the endpoint at the follow-up visit concerned.

## Analysis of forecast errors

We analyzed the distribution of mean absolute errors with a mixed-effects model. We corrected the errors for several possible cofounding factors and accounted for the fact that multiple forecasts originated from the same participant. In practice, for a given endpoint and a given model, we performed the following procedure 100 times:

- We randomly picked a subset of disjointed prediction tasks, namely predictions not sharing any common visit (neither the blinded visit to forecast, nor the unblinded visits used to forecast);

- We fit a multivariate linear mixed-effects model with a random intercept for each individual, using the following categorical explanatory variables: A/T/N/C stage at prediction, cohort, number of APOE-ε4 alleles, sex, level of education, number of unblinded visits, and continuous explanatory variables: actual patient's age at prediction centered on 75 years and normalized by 7.5 years, years to prediction centered on three years and normalized by one year, mean time between unblinded visits centered on eight months and normalized by three months, percentage of missing data for the unblinded visits per modality.

Education level was classified as low if the subject had followed no more than nine years of formal education and high if the subject had followed at least 16 years of education, in accordance with the guidelines of the international standard classification of education of the United Nations.

We derived the mean and empirical confidence interval for the model intercept (the mean absolute error adjusted for cofounding factors) and regression coefficients (association between the mean absolute errors and each cofounding factor).

## Comparison with alternative methods

We also compared AD Course Map with two additional alternative methods. The first, the no-change prediction or last-observation-carried-forward method, forecasts the future value of an endpoint to be the same as it was at the last unblinded visit. The second method, the linear mixed model method, involved generating a linear mixed-effects model for each endpoint, regressing endpoint values against the age of the participant at the successive visits, with a random intercept and a random slope per subject. The model was fitted to an unseen participant with a maximum a posteriori estimator[67]. We used the same validation procedure for all models: AD Course Map, RNN-AD, no-change prediction, and the linear mixed model.

## Clinical trial simulation, enrichment evaluation, and sample size calculation

We simulated clinical trials in subjects at risk of developing AD or at an early stage of AD, as described in Table 3. For each trial, we selected all pairs of visits from all participants in the five data sets satisfying the following criteria:

- The primary endpoint of the trial was assessed at both visits,
- The patient fulfilled the inclusion criteria and had none of the exclusion criteria of the trial at the baseline visit,

- Visits were separated by the duration of the trial, with a certain tolerance, depending on the trial.

For each pair of visits, the first was considered to be the baseline visit at inclusion and the second was considered to be the visit at the end of the trial. We did not take into account possible intermediate visits. Supplementary Table 8 summarizes the characteristics of participants included in all the simulated trials.

We first evaluated our prognostic enrichment strategy from a diagnostic test standpoint. For each trial, we forecast the value of the primary endpoint at the follow-up visit from the baseline data only, using the procedure described above. We calculated the median value of the outcome (i.e. the annual rate of change between baseline and follow-up visit). Participants above this threshold were considered to be fast progressors and formed the target population to be identified. A threshold for predicted outcomes was used to split the population into two groups: one considered at high risk of progression and the other at low risk of progression. We let the low-risk vs. high-risk threshold vary and calculated the resulting receiver operator characteristic (ROC) curve. On this curve, we identified the point splitting the population into a low-risk and a high-risk group of equal sizes, which was used as the operating point. We determined confidence intervals by performing our analyses 100 times on half the samples selected at random. Within any given run, any visit of a patient was used no more than once. The regions of confidence around ROC curves were constructed graphically as envelopes of both sensitivity and specificity confidence intervals along thresholds.

We evaluated possible biases in the group at high risk of progression. We used a logistic regression predicting selection status from population covariates (age, sex, education, number of APOE-ε4 alleles), cohort, and missing baseline modalities, together with the true indicator of fast progression. This last binary predictor was included to check for biases emerging in addition to the biases naturally present in the target population.

We then evaluated our prognostic enrichment strategy by calculating statistical power. We used a hypothetical individual treatment model: if the outcome actually worsened between baseline and follow-up for the participant, we changed the annual rate of change by the treatment effect, e.g. a 20% improvement of the annual rate of change. We did not apply a treatment effect if the participant improved between baseline and follow-up. For treatment effects ranging from 20% to 30%, we computed effect size (Cohen's $d$) and sample size from a two-independent sample asymptotic $t$-test, with a 5% bilateral level of significance and 80% statistical power. We compared this sample size for the population selected with the trial inclusion criteria alone, and for the subpopulation identified as at high risk of progression. We reported the total sample size for two arms. We did not account for the drop-out rate in the calculation, as the goal was to compare statistical power with and without enrichment.

In these two experiments, we compared the results obtained with those for a method selecting APOE-ε4 carriers (heterozygous or homozygous) as participants at high risk of progression. We were unable to use this method for the trial targeting participants at risk of the onset of AD since this trial included only APOE-ε4 carriers.

## Statistics and reproducibility

No statistical method was used to predetermine the sample size. We considered all available data from all the cohorts and excluded only the data of the participants with less than one year of follow-up. The experiments were not randomized since only observational data were used. The investigators were not blinded to allocation during experiments and outcome assessment since only observational data were used. Simulations of clinical trials included a random unblinded allocation into treated and control arms with assessment of biases in sex, center, level of education, and APOE genotype.

## Reporting summary

Further information on research design is available in the Nature Portfolio Reporting Summary linked to this article.

## Data availability

The ADNI and AIBL data used in this study are available in the database of the laboratory of neuroimaging at the University of Southern California under accession code at http://adni.loni.usc.edu. The J-ADNI data used in this study are available in the NBDC Human Database under accession code at http://humandbs.biosciencedbc.jp/en/. The PharmaCog data used in this study are available in the NeuGRID2 platform under access code at https://www.neugrid2.eu/ (https://doi.org/10.17616/R31NJN1E). The MEMENTO data used in this study are available in Dementia Platform UK under accession code at https://portal.dementiasplatform.uk/CohortDirectory/Item?fingerPrintID=MEMENTO. Raw data and patient-level data that were generated in this study are protected and are not available due to data privacy laws and data use agreements. These data can be re-generated using the open-source software Leaspy (see below) by anyone with authorized access to the above third-party data. The data used to compute the statistics in this study are available in a dedicated Zenodo repository[68]. Source data are provided with this paper.

## Code availability

The statistical analysis of the forecast errors and the simulation of clinical trials were performed in Python. We used the Leaspy open-source software (https://gitlab.com/icm-institute/aramislab/leaspy) for training and testing AD Course Map, and the corresponding open-source software for RNN-AD https://github.com/ThomasYeoLab/CBIG/tree/master/stable_projects/predict_phenotypes/Nguyen2020_RNNAD. Linear mixed models were trained using the open-source *statsmodels* package[69]. The frozen versions of the Python libraries that were used to generate the results of this article can be found in a dedicated Zenodo repository[68].

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

## Acknowledgements

E.M., I.K., J.O., S.E., and S.D. received funding from the European Research Council (ERC) under grant agreement no. 678304, the Agence Nationale de la Recherche (ANR) under the "Investissements d'avenir" program, grants ANR-10-IAIHU-06 (IHU ICM) and ANR-19-P3IA-0001 (PRAIRIE 3IA Institute). E.M., I.K., J.O., C.B., M.H.A., and S.D. received funding from the European Union's Horizon 2020 research and innovation program under grant agreement no. 826421 (TVB-Cloud). D.A. and S.D. received funding from the joint program in neurodegenerative diseases (JPND) under grant agreement ANR-19-JPW2-000 (E-DADS). S.E. received funding via an APHP-Inria collaboration grant (poste d'accueil) within the ARAMIS project-team. The data used in the preparation of this article were obtained from:

- The Alzheimer's Disease Neuroimaging Initiative (ADNI) database (adni.loni.usc.edu),
- The Australian Imaging Biomarkers and Lifestyle flagship study of ageing (AIBL),
- The Japanese Alzheimer's Disease Neuroimaging Initiative (J-ADNI),
- The PharmaCog Consortium,
- The MEMENTO cohort.

The investigators of these studies contributed to the design and implementation of the corresponding cohorts, provided data, but did not participate in analysis or writing of this report. Complete listings of these investigators can be found at:

- ADNI: https://adni.loni.usc.edu/wp-content/uploads/how_to_apply/ADNI_Acknowledgement_List.pdf
- AIBL: https://aibl.csiro.au/about/aibl-research-team/
- J-ADNI: https://humandbs.biosciencedbc.jp/en/hum0043-j-adni-authors
- PharmaCog: https://neugrid2.eu/wp-content/uploads/2022/01/pharmacog_investigators_list.pdf,
- MEMENTO: supplementary material.

Data collection and sharing were partly funded by the ADNI (National Institutes of Health Grant U01 AG024904) and DOD ADNI (Department of Defense award number W81XWH-12-2-0012). ADNI is funded by the National Institute on Aging, the National Institute of Biomedical Imaging and Bioengineering, and through generous contributions from the following: AbbVie, Alzheimer's Association; Alzheimer's Drug Discovery Foundation; Araclon Biotech; BioClinica, Inc.; Biogen; Bristol-Myers Squibb Company; CereSpir, Inc.; Cogstate; Eisai Inc.; Elan Pharmaceuticals, Inc.; Eli Lilly and Company; EuroImmun; F. Hoffmann-La Roche Ltd and its affiliated company Genentech, Inc.; Fujirebio; GE Healthcare; IXICO Ltd.; Janssen Alzheimer Immunotherapy Research & Development, LLC.; Johnson & Johnson Pharmaceutical Research & Development LLC.; Lumosity; Lundbeck; Merck & Co., Inc.; Meso Scale Diagnostics, LLC.; NeuroRx Research; Neurotrack Technologies; Novartis Pharmaceuticals Corporation; Pfizer Inc.; Piramal Imaging; Servier; Takeda Pharmaceutical Company; and Transition Therapeutics. The Canadian Institutes of Health Research is providing funds to support ADNI clinical sites in Canada. Private-sector contributions are facilitated by the Foundation for the National Institutes of Health (www.fnih.org). The grantee organization is the Northern California Institute for Research and Education, and the study is coordinated by the Alzheimer's Therapeutic Research Institute at the University of Southern California. ADNI data are disseminated by the Laboratory for Neuroimaging at the University of Southern California. AIBL core funding was provided by the Commonwealth Scientific and Industrial Research Organization

(CSIRO). The cohort was also supported by the University of Melbourne, Neurosciences Australia Ltd, Edith Cowan University, Mental Health Research Institute, Alzheimer's Australia, National Ageing Research Institute, Austin Health, University of WA, CogState Ltd., Macquarie University, Hollywood Private Hospital, Sir Charles Gairdner Hospital. J-ADNI was supported by the following grants: Translational Research Promotion Project from the New Energy and Industrial Technology Development Organization of Japan; Research on Dementia, Health Labor Sciences Research Grant; Life Science Database Integration Project of Japan Science and Technology Agency; Research Association of Biotechnology (contributed by Astellas Pharma Inc., Bristol-Myers Squibb, Daiichi-Sankyo, Eisai, Eli Lilly and Company, Merck-Banyu, Mitsubishi Tanabe Pharma, Pfizer Inc., Shionogi & Co., Ltd., Sumitomo Dainippon, and Takeda Pharmaceutical Company), Japan, and a grant from an anonymous foundation. The IMI-PharmaCog/E-ADNI project was funded by the European seventh framework program and European Federation of Pharmaceutical Industries and Associations (EFPIA) for the Innovative Medicine Initiative (Grant no. 115009; http://www.pharmacog.org). The MEMENTO cohort was funded through research grants from the Fondation Plan Alzheimer (Alzheimer Plan 2008–2012), and the French Ministry of Higher Education, Research and Innovation (Plan Maladies Neurodégénératives 2014-2019). This work was also supported by CIC1401-EC, Bordeaux University Hospital (CHU Bordeaux, sponsor of the cohort), Inserm, and the University of Bordeaux. The MEMENTO cohort has received funding from AVID, GE Healthcare, and FUJIREBIO through private-public partnerships. This work was undertaken with resources from the Dementias Platform UK (DPUK) Data Portal; the Medical Research Council supports DPUK through grant MR/L023784/2.

## Author contributions

S.E., C.D., M.H.-A., and S.D. conceived and supervised the research. E.M., I.K., J.O., C.B., D.A., V.B. conducted the research and performed data analysis. All authors contributed to writing the paper and agreed on its content. Editorial support, in the form of medical writing and copy-editing, was provided by Julie Sappa of Alex Edelman and Associates.

## Competing interests

S.E. received personal fees from Biogen, Eisai, Roche, and GE Healthcare for presentations and participation in advisory boards. S.D. is co-inventor of the patent "a method for determining the temporal progression of a biological phenomenon and associated methods and devices" which protects the potential uses of AD Course Map including targeting the time to administer medicine or identifying a biomarker (applicants: Inserm, CNRS, Sorbonne Université, Inria, Ecole Polytechnique, ICM, AP-HP, inventors: Stanley Durrleman, Jean-Baptiste Schiratti, Stéphanie Allassonnière, Olivier Colliot, international application number: PCT/IB2016/052699, granted in the USA (grant number 10832089), published in Europe and Japan. The remaining authors declare no competing interests.
