## [Peer Review File · Nature Communications]

Forecasting Individual Progression Trajectories in Alzheimer's DiseaseReviewer #1 (Remarks to the Author):

The authors use a disease progression model to map out the time course of biomarker changes as a function of age. A progression model uses a series of logistic curves going. The model includes parameters such as a time shift to fit the shape and position of the curves. Two models were fit with slightly different data inclusion criteria and fit on the ADI data. Using these models the authors predict future data. They compare the error estimates from their prediction to a two models; a no-change model and a model that assumes a linear change as a function of time. When examining the clinical trial simulation the authors split individuals into "fast" and "slow" progressors based upon a median value. They then estimate gains in power if only the optimal group was selected. The general questions of the paper are good and interesting to the field. The paper has somewhat of an identity crisis and is split into two halves. The first part is describing the model, while the second is on applying the model and how this may impact trials. The paper is very methods heavy, and as a result isn't ideal for the targeted journal. I felt at times that there simply wasn't enough explanation of what is going on. As a result the paper often had a black box feel to me.

Comments

1. The training set uses the 823 amyloid positive ADNI participants. The authors indicate that they used the other ADNI participants as one of the five test sets. It would seem problematic to have an AD prediction model on individuals that you are a priori screening out to not have AD pathology. Later on in the methods the authors indicate that they determined the threshold for be the value in the logistic regression that best discriminated amyloid positive ADNI participants from the rest of the cohort. So were the amyloid negative individuals used when determining the curves or not? When hitting the second of the forecasting endpoints it is the held out 20% of amyloid positive individuals and the rest of the cohort that are used. Again, it seems meaningless to predict clinical progression using AD biomarkers in individuals free of such pathology.

2. The forecasting of data points is interesting, but the paper could do a better job describing that longitudinal data is present. Right now there is just the table. It would be useful to have a bit of this data in the methods text as well (n visits, duration of follow-up). Were all modalities present at all visits, or are some data points only there at a subset of visits (e.g. tau PET was added later in ADNI so you don't have 17 years of follow-up). This would make it easier on the reader to grasp what you are doing without having to jump around. This is a personal preference though more than anything.

There were other parts of this section that just didn't make sense to me. In the methods you state "with a follow-up visit between 1.4 and 6.6 months after the last known visit." Do you mean follow-up after the baseline visit? How can you verify a prediction error in a time point after the last known visit (i.e. no acquired data)?

You then examine the forecasting error using a LME because "multiple forecasts originated from the same participant." Why is this occurring? Is this because you have multiple follow-up visits and you are looking at the error at each one? Since this is the key element of the paper please read through this section and make it as clear as possible.

3. How does the model handle missing data? Was there a threshold of complete data needed for inclusion?

4. The authors indicate that measures were "normalized" between 0 and 1. Typically when I think of normalization I think of a transformation to z-scores. Instead you are using a Box-Cox transformation and then a linear rescaling to make things 0 to 1. This approach assumes that the clinical characteristics of your different cohorts are similar. This is a strong assumption. Did the authors make any attempt to justify this assumption? Was this simply done as a limitation of the progression model?

I tend to not like setting the boundaries of all biomarkers to be the same as this is not how the biomarkers actually behave in the disease. The degree of abnormality for say AB42 and amyloid PET are orders of magnitude larger the deviance in hippocampal volume relative to individuals without AD. As a result using such an approach distorts the actual pattern of abnormality. It would be better to use a reference cohort and calculate actual z-scores. In this way you still get values onto the same scale, but you preserve the fact that different markers show variable levels of divergence from normality. This may though be a limitation of your progression model.

5. Combining CSF data from multiple cohorts is quite difficult. This is due to the wildly different ranges that can be produced across assays as well as the fact that there is substantial drift over time in AB assays (Bijms et al., 2018, Schindler et al., 2018). This drift is likely less of an issue in the ADNI and J-ADNI data but will be an issue for AIBL, PharmaCog, and Memento data. How was variability across assay lots accounted for?

6. In the paper the authors indicate they are using a whole-brain SUVR for both AV45 and AV1451 but the methods make it clear that this isn't the case. Update the methods early in the paper to make it clear that you are using tau specific summary.

7. The authors indicate that they cannot combine amyloid PET data between cohorts. There should be Centiloid equations available for both ADNI and AIBL. I'm unsure about Memento. Is PET data not available for J-ADNI or PharmaCog? Is tau PET only coming from ADNI?

8. The authors indicate that they calculated two different progression models. When reading the introduction the rationale for this wasn't clear. When I jumped to the methods it appears likely that this is done because PET imaging is not present for all of the cohorts. It was never clear to me why you were running these two parallel models that are near identical. It also isn't clear unless you go down to the methods or table that the five cohorts don't all contain PET.

9. The raw values for the CSF biomarkers across the cohorts will vary as a function of the platform. You may think about denoting this somehow visually in the table to avoid confusion.

10. In the results the authors say "AD Course Map therefore predicts the subject-specific trajectory of biomarker changes from data collected from the subject." From my reading the Course Map is fit using the aggregate group data. How is it taking into account the subject-specific factors? The authors also indicated that the prediction models are taking into account additional historical data? How is this being done? While the general idea of the model is laid out, there is a fair amount that is coming off to me as opaque. Please make what you are doing as transparent as possible.

11. The CSF AB42, ptau181, and amyloid PET values start out very abnormal. This is likely a byproduct of constraining the dataset to amyloid positive individuals? This however does highlight my concern in point #4. In AD the degree of abnormality for amyloid and tau markers far outpaces that of any other biomarker. In your figures though volumetrics, which are far less affected, reach a higher level of abnormality. This is almost certainly due to the fact you are fixing the maximum degree of abnormality across all biomarkers to the same maximal value. As a result your model is distorting the relative degree of abnormality of the different markers.

12. In Figure 3 the authors indicate that their models are better than alternate models. One of those models is a no-change model while the other constrains longitudinal change to be linear. We know that both of those alternate models are almost certainly incorrect beforehand. We know biomarker and cognitive data change. We also know that they typically behave in a non-linear fashion. These comparisons may be an okay qualitative check for your model, but they are not particularly robust comparisons. I would not overstate this result. The improvement may be less from your particular

model and more in the weakness of the assumptions in your alternate.

13. I have somewhat of a similar concern with the data presented in Figure 5. I think this is fine descriptive data, but there is no comparative data so it is hard to know what to make of the ROC curves. It is difficult to judge of the prediction rates you are getting are good or bad. The marker of the E4 is helpful, but only one isolated point.

14. Figures 5 and 6 need better context. A reader is not going to want to jump back and forth between the figure and the table to figure out what the predicted measure is.

15. In the discussion you state In this work "we compared theoretically treated participants with a digital "twin" displaying natural disease progression. This approach makes it possible to generate a synthetic placebo participant exactly matching the treated participants automatically. This approach makes it possible to generate a synthetic placebo participant exactly matching the treated participants automatically. This application of our method would further limit the number of participants in the placebo arms of trials, provided that a placebo effect is included." I wouldn't say this. Your synthetic placebo is just a model prediction. An error in a model prediction would show up as a "treatment" effect in the absence of any real drug effect. Your comparison is using data from real people, not from digital twins. It is very unlikely that this sort of thing would be acceptable to any regulatory agency in the world. I'd recommend striking this assertion from your discussion.

16. I would not say that your model is better than 56 alternative models. None of that data is presented in the current manuscript

17. Figure 6 is probably the most important element of your paper. While there is a description of what you are doing it really felt to me very much like a black box. For what is in essence really a methods heavy paper the format you are submitting to leads to a very brief communication of what you are doing. This is unfortunate.

Reviewer #2 (Remarks to the Author):

This impressive work evaluates a statistical model, AD Course Map, for prediction of progression in patients with (suspected) Alzheimer's disease in data from five longitudinal multicenter cohorts (total of 4634 participants).

The aim of AD Course Map is "prognostic enrichment" of cohorts for clinical trials, i.e. select those patients that are likely to display progression during the trial.

This paper validated that aim by simulating clinical trials (based on characteristics of real clinical trials) from the longitudinal cohorts. They evaluated predictive performance of trials' primary outcomes (cognition) as well as expected sample size reduction. Also bias, generalizability and robustness to missing data was analyzed.

I find the paper and the presented analysis very strong. The authors did a lot of work and managed present this in a very clear, novel and convincing way. The paper presents a lot of detail in a very comprehensive way. The analysis is done in a very thorough way. The results are very relevant and important for setting up future clinical trials for AD.

Major:

- How were the two sets of end points for AD Course Maps selected (line 362)? I would understand the choice of comparing a PET versus a non-PET model (given that PET is expensive and not widely available yet), but miss an explanation for using CBR-SB in the non-PET but not the PET model.

- The clinical trial simulation explanation would benefit from a figure visually showing the steps in the simulation and analysis. (Like figure 1, but then for the simulation

experiment). Especially the 'hypothetical treatment' gets a bit lost in the current form and took me a while to understand.

Minor:

- The explanation of AD Course Map is very brief and cannot be understood with reading a bit of Ref 6. Please expand a bit in this paper.
- Line 239: The authors that the method is fair as there are no biases due to sex. Please explain how this is ensured, by using balanced training data?
- Figure 3, CDR-SB: I assume the last box in the last subplot should be red instead of green (PET-model predicting CBR-SB). Please correct or clarify.
- Figure 5: I took me quite some time to fully understand the subplots and their relation to their titles. It would help to include the target variable (MMSE, ADAS, CDR) for each subplot between brackets. And to change the second sentence to something like "Receiver operating characteristic (ROC) curves are shown for the six simulated trials showing the performance of AD course map in selecting the group of participants with the largest change in primary outcome during follow-up."

Reviewer #3 (Remarks to the Author):

The manuscript entitled "FORECASTING INDIVIDUAL PROGRESSION TRAJECTORIES FOR PROGNOSTIC ENRICHMENT IN ALZHEIMER'S DISEASE TRIALS" looks at five different cohorts of Alzheimer's disease patients to build a map of how different variables related to AD pathology or demographic and clinical factors worsen over time. The resulting model from this longitudinal map is used to forecast follow up visit outcomes in patients. Building on this, a model-based enrichment strategy is proposed and simulations are used to estimate potential benefits in increasing statistical power in hypothetical trials. The manuscript is well written and easy to read. The concepts are very novel and not encountered in prior literature. The advent of new treatments call for exactly the kind of strategies explained here and in my opinion the perceived impact of this paper is high for the field in developing new treatments in AD in the near future. The sample size is large, multi-center and encompasses a diverse set of populations. Overall the methods are solid and have been developed through several prior projects of the participating research groups. My biggest concern about this study is the lack of a real external cohort. While there are more than five cohorts, none is set outside of modeling and training, and as explained in Methods (although difficult to understand), it seems that only N-fold cross-validation is used there. Specifically, in the sample size and power analysis (enrichment part, end of Methods): it is unclear what model (from many cross-validation folds) is used in forecasting (it is mentioned test subjects were used, but since there is no test dataset defined a priori or clearly to the reader, it leaves the impression that patients from the same cohort are used in the assessment of models that is not ideal as the gold standard in machine learning literature). The manuscript leaves the impression that there was no a priori, external data set for testing the forecasts. There are some unexpected and odd aspects to reporting results (I have specified below, e.g., using 90% CI), with significant prediction/forecast uncertainty (some of which make the observations less relevant to practice, if at all). My specific comments are below:

- Unclear why confidence intervals are used in a Bayesian paradigm.
- The manuscript should change 90% CI to 95% CI. It seems to the reviewer that if this happens at all (page 6, starting from line 127), the reported biomarker trajectories become non-significant. A major problem in this model is the very wide intervals and uncertainty. It seems that the model has failed to capture variability, and it is unclear why the normal 95% CI convention is changed to 90% CI (this should have been done a priori and with a detailed explanation). The manuscript seems to be selectively reporting 95% CI for other measures.
- The use of the term 'calibration' is vague in Methods. The manuscript probably is referring to some sort of variable adjustment but is not explained. Remove this term and

explain specifically what procedure is done to adjust/calibrate models

-Why was the whole brain SUVR used for modeling, while regional atrophy (hippocampus) was used from MRI?

-Choice of linear mixed-effects models and 'no change' models as two benchmarks for the presented model in enrichment is very modest. I suggest using more powerful models designed for non-linear predictions to show to readers how powerful the proposed model is (I can think of deep learning survival models or similar, my point is that the manuscript is not using a challenging benchmark).

-I suggest using the continuous SUVR measures for amyloid-beta, even if different tracers are used. Dichotomizing discards significant useful information in such a rich and large cohort of patients.

-I also find the division of patients into slow/fast progressors problematic and not the best practice. As I mentioned above, using a forecast model, a continuous or probabilistic outcome prediction should be used to enrich the trial. The manuscript seems to be going back and forth between continuous outcomes / mean square error and AUC/ dichotomization (this is very confusing for the reader). Similar to the point above, I believe lots of important data and nuances of modeling are lost because of this thresholding.

Arman Eshaghi

Dear reviewers,

We would like to warmly thank you for your careful read, comments, and relevant suggestions regarding our manuscript.

We revised our work to address three major concerns regarding benchmark with competing methods, use of external validation cohorts and data normalization.

Regarding the benchmark, our first submission compared the performance of AD Course Map with two alternative methods: a no-change scenario and linear mixed-effect model. In this revision, we also compared AD Course Map with RNN-AD, a recurrent neural network aiming, like AD Course Map, to predict the changes of endpoints during AD progression. RNN-AD was developed by an external research laboratory. We used the software code that was publicly released by its authors. RNN-AD ranked #2 in the TADPOLE challenge in 2019. It provides therefore a fair and challenging alternative to AD Course Map. We show in this revised version of the manuscript that, although RNN-AD also outperformed the no-change scenario and linear mixed model, AD Course Map provides better and more robust performance for forecasting disease progression, selecting fast progressors and decreasing sample size in trials compared to RNN-AD.

We also took advantage of the addition of RNN-AD as an alternative to clarify the core contribution of our work: the use of disease progression models to better select participants in trials. We removed the illustration and interpretation of AD Course Map (previous Figure 2 showing the typical changes of the endpoints during AD progression). They added more confusion than clarification. We replace this figure by an illustration of how AD Course Map is used to forecast endpoints values (now Figure 1), and another figure to illustrate how these forecasts serve as enrichment criteria in trials (now Figure 4). We hope that these changes will address R1's concern about the "identity crisis" of the article.

Regarding the use of external validation cohorts, it is very unfortunate that we did not explain more clearly that 4 out of the 5 cohorts used in this analysis were indeed external validation cohorts. Models were trained using only a subset of ADNI. The rest of the ADNI participants and all participants from the 4 other cohorts (J-ADNI, AIBL, PharmaCog, and Memento) were used only at test time for the evaluation of the forecast errors, the classification metrics of slow vs fast progressors and the sample size computation. No training data were used for these tests. We made our best efforts to clarify our validation procedure in this revision of the manuscript. We also paid a particular attention to use a consistent terminology throughout the text (e.g. using "training" instead of "calibration").

Regarding data normalization, we added more detailed explanations in the Methods section about how we harmonized the data across cohorts and normalized them for use as input to the model. Harmonization ensures that CSF data with different immuno-assays and amyloid PET data with different radiotracers are comparable. For CSF data, we centered the measurements and reduced their variance to 1 after controlling for age, APOE genotype and clinical dementia rating scale within each cohort. For amyloid PET data, we converted the measurements in centiloid scale by using equations provided in the literature, as suggested by reviewers. These new harmonization procedures changed results only marginally compared to the first version of the manuscript. The normalization into a unit scale is an internal step within each method. AD Course Map and RNN-AD have their own internal procedure: AD Course Map converts data using a clipped linear transform, RNN-AD converts data into z-scores. In any case, the output of the model is always converted back to the native scale (and unit) of the measurement before being analyzed. A bad normalization procedure could only decrease the performance of the method.

In addition to the specific points discussed above, we also revised the manuscript to account for all your other comments. You will find below a point-by-point response. Your comments are in italics, followed by our response and, when appropriate, the changes we made in the text are in bold.

We hope that our additional analyses together with the revision of the manuscript will address the reservations you had about our article.

Reports on the manuscript

Reviewer #1 (Remarks to the Author):

The authors use a disease progression model to map out the time course of biomarker changes as a function of age. A progression model uses a series of logistic curves going. The model includes parameters such as a time shift to fit the shape and position of the curves. Two models were fit with slightly different data inclusion criteria and fit on the ADI data. Using these models the authors predict future data. They compare the error estimates from their prediction to a two models; a no-change model and a model that assumes a linear change as a function of time. When examining the clinical trial simulation the authors split individuals into “fast” and “slow” progressors based upon a median value. They then estimate gains in power if only the optimal group was selected.

The general questions of the paper are good and interesting to the field. The paper has somewhat of an identity crisis and is split into two halves. The first part is describing the model, while the second is on applying the model and how this may impact trials. The paper is very methods heavy, and as a result isn't ideal for the targeted journal. I felt at times that there simply wasn't enough explanation of what is going on. As a result the paper often had a black box feel to me.

Comments

1. The training set uses the 823 amyloid positive ADNI participants. The authors indicate that they used the other ADNI participants as one of the five test sets. It would seem problematic to have an AD prediction model on individuals that you are a priori screening out to not have AD pathology. Later on in the methods the authors indicate that they determined the threshold for be the value in the logistic regression that best discriminated amyloid positive ADNI participants from the rest of the cohort. So were the amyloid negative individuals used when determining the curves or not? When hitting the second of the forecasting endpoints it is the held out 20% of amyloid positive individuals and the rest of the cohort that are used. Again, it seems meaningless to predict clinical progression using AD biomarkers in individuals free of such pathology.

2. The forecasting of data points is interesting, but the paper could do a better job describing that longitudinal data is present. Right now there is just the table. It would be useful to have a bit of this data in the methods text as well (n visits, duration of follow-up). Were all modalities present at all visits, or are some data points only there at a subset of visits (e.g. tau PET was added later in ADNI so you don't have 17 years of follow-up). This would make it easier on the reader to grasp what you are doing without having to jump around. This is a personal preference though more than anything.

There were other parts of this section that just didn't make sense to me. In the methods you state “with a follow-up visit between 1.4 and 6.6 months after the last known visit.” Do you mean follow-up after the baseline visit? How can you verify a prediction error in a time point after the last known visit (i.e. no acquired data)?

You then examine the forecasting error using a LME because “multiple forecasts originated from the same participant.” Why is this occurring? Is this because you have multiple follow-up visits and you are looking at the error at each one? Since this is the key element of the paper please read through this section and make it as clear as possible.

3. How does the model handle missing data? Was there a threshold of complete data needed for inclusion?

4. The authors indicate that measures were “normalized” between 0 and 1. Typically when I think of normalization I think of a transformation to z-scores. Instead you are using a Box-Cox transformation and then a linear rescaling to make things 0 to 1. This approach assumes that the clinical characteristics of your different cohorts are similar. This is a strong assumption. Did the authors make any attempt to justify this assumption? Was this simply done as a limitation of the progression model?

I tend to not like setting the boundaries of all biomarkers to be the same as this is not how the biomarkers actually behave in the disease. The degree of abnormality for say AB42 and amyloid PET are orders of magnitude larger the deviance in hippocampal volume relative to individuals without AD. As a result using such an approach distorts the actual pattern of abnormality. It would be better to use a reference cohort and calculate actual z-scores. In this way you still get values onto the same scale, but you preserve the fact that different markers show variable levels of divergence from normality. This may though be a limitation of your progression model.

5. Combining CSF data from multiple cohorts is quite difficult. This is due to the wildly different ranges that can be produced across assays as well as the fact that there is substantial drift over time in AB assays (Bijms et al., 2018, Schindler et al., 2018). This drift is likely less of an issue in the ADNI and J-ADNI data but will be an issue for AIBL, PharmaCog, and Memento data. How was variability across assay lots accounted for?

6. In the paper the authors indicate they are using a whole-brain SUVR for both AV45 and AV1451 but the methods make it clear that this isn't the case. Update the methods early in the paper to make it clear that you are using tau specific summary.

7. The authors indicate that they cannot combine amyloid PET data between cohorts. There should be Centiloid equations available for both ADNI and AIBL. I'm unsure about Memento. Is PET data not available for J-ADNI or PharmaCog? Is tau PET only coming from ADNI?

8. The authors indicate that they calculated two different progression models. When reading the introduction the rationale for this wasn't clear. When I jumped to the methods it appears likely that this is done because PET imaging is not present for all of the cohorts. It was never clear to me why you were running these two parallel models that are near identical. It also isn't clear unless you go down to the methods or table that the five cohorts don't all contain PET.

9. The raw values for the CSF biomarkers across the cohorts will vary as a function of the platform. You may think about denoting this somehow visually in the table to avoid confusion.

10. In the results the authors say "AD Course Map therefore predicts the subject-specific trajectory of biomarker changes from data collected from the subject." From my reading the Course Map is fit using the aggregate group data. How is it taking into account the subject-specific factors? The authors also indicated that the prediction models are taking into account additional historical data? How is this being done? While the general idea of the model is laid out, there is a fair amount that is coming off to me as opaque. Please make what you are doing as transparent as possible.

11. The CSF AB42, ptau181, and amyloid PET values start out very abnormal. This is likely a byproduct of constraining the dataset to amyloid positive individuals? This however does highlight my concern in point #4. In AD the degree of abnormality for amyloid and tau markers far outpaces that of any other biomarker. In your figures though volumetrics, which are far less affected, reach a higher level of abnormality. This is almost certainly due to the fact you are fixing the maximum degree of abnormality across all biomarkers to the same maximal value. As a result your model is distorting the relative degree of abnormality of the different markers.

12. In Figure 3 the authors indicate that their models are better than alternate models. One of those models is a no-change model while the other constrains longitudinal change to be linear. We know that both of those alternate models are almost certainly incorrect beforehand. We know biomarker and cognitive data change. We also know that they typically behave in a non-linear fashion. These comparisons may be an okay qualitative check for your model, but they are not particularly robust comparisons. I would not overstate this result. The improvement may be less from your particular model and more in the weakness of the assumptions in your alternate.

13. I have somewhat of a similar concern with the data presented in Figure 5. I think this is fine descriptive data, but there is no comparative data so it is hard to know what to make of the ROC curves. It is difficult to judge of the prediction rates you are getting are good or bad. The marker of the E4 is helpful, but only one isolated point.

14. Figures 5 and 6 need better context. A reader is not going to want to jump back and forth between the figure and the table to figure out what the predicted measure is.

15. In the discussion you state In this work "we compared theoretically treated participants with a digital "twin" displaying natural disease progression. This approach makes it possible to generate a synthetic placebo participant exactly matching the treated participants automatically. This approach makes it possible to generate a synthetic placebo participant exactly matching the treated participants automatically. This application of our method would further limit the number of participants in the placebo arms of trials, provided that a placebo effect is included." I wouldn't say this. Your synthetic placebo is just a model prediction. An error in a model prediction would show up as a "treatment" effect in the absence of any real drug effect. Your comparison is using data from real people, not from digital twins. It is very unlikely that this sort of thing would be acceptable to any regulatory agency in the world. I'd recommend striking this assertion from your discussion.

16. I would not say that your model is better than 56 alternative models. None of that data is presented in the current manuscript

17. Figure 6 is probably the most important element of your paper. While there is a description of what you are doing it really felt to me very much like a black box. For what is in essence really a methods heavy paper the format you are submitting to leads to a very brief communication of what you are doing. This is unfortunate.

Reviewer #2 (Remarks to the Author):

This impressive work evaluates a statistical model, AD Course Map, for prediction of progression in patients with (suspected) Alzheimer's disease in data from five longitudinal multicenter cohorts (total of 4634 participants).

The aim of AD Course Map is "prognostic enrichment" of cohorts for clinical trials, i.e. select those patients that are likely to display progression during the trial.

This paper validated that aim by simulating clinical trials (based on characteristics of real clinical trials) from the longitudinal cohorts. They evaluated predictive performance of trials' primary outcomes (cognition) as well as expected sample size reduction. Also bias, generalizability and robustness to missing data was analyzed.

I find the paper and the presented analysis very strong. The authors did a lot of work and managed present this is a very clear, novel and convincing way. The paper present a lot of detail in a very comprehensive way. The analysis is done in a very thorough way. The results are very relevant and important for setting up future clinical trials for AD.

Major:

- How were the two sets of end points for AD Course Maps selected (line 362)? I would understand the choice of comparing a PET versus a non-PET model (given that PET is expensive and not widely available yet), but miss an explanation for using CBR-SB in the non-PET but not the PET model.
- The clinical trial simulation explanation would benefit from a figure visually showing the steps in the simulation and analysis. (Like figure 1, but then for the simulation experiment). Especially the 'hypothetical treatment' gets a bit lost in the current form and took me a while to understand.

Minor:

- The explanation of AD Course Map is very brief and cannot be understood with reading a bit of Ref 6. Please expand a bit in this paper.
- Line 239: The authors that the method is fair as there are no biases due to sex. Please explain how this is ensured, by using balanced training data?
- Figure 3, CDR-SB: I assume the last box in the last subplot should be red instead of green (PET-model predicting CBR-SB). Please correct or clarify.
- Figure 5: I took me quite some time to fully understand the subplots and their relation to their titles. It would help to include the target variable (MMSE, ADAS, CDR) for each subplot between brackets. And to change the second sentence to something like "Receiver operating characteristic (ROC) curves are shown for the six simulated trials showing the performance of AD course map in selecting the group of participants with the largest change in primary outcome during follow-up."

Reviewer #3 (Remarks to the Author):

The manuscript entitled "FORECASTING INDIVIDUAL PROGRESSION TRAJECTORIES FOR PROGNOSTIC ENRICHMENT IN ALZHEIMER'S DISEASE TRIALS" looks at five different cohorts of Alzheimer's disease patients to build a map of how different variables related to AD pathology or demographic and clinical factors worsen over time. The resulting model from this longitudinal map is used to forecast follow up visit outcomes in patients. Building on this, a model-based enrichment strategy is proposed and simulations are used to estimate potential benefits in increasing statistical power in hypothetical trials. The manuscript is well written and easy to read. The concepts are very novel and not encountered in prior literature. The advent of new treatments call for exactly the kind of strategies explained here and in my opinion the perceived impact of this paper is high for the field in developing new treatments in AD in the near future. The sample size is large, multi-center and encompasses a diverse set of populations. Overall the methods are solid and have been developed

through several prior projects of the participating research groups. My biggest concern about this study is the lack of a real external cohort. While there are more than five cohorts, none is set outside of modeling and training, and as explained in Methods (although difficult to understand), it seems that only N-fold cross-validation is used there. Specifically, in the sample size and power analysis (enrichment part, end of Methods): it is unclear what model (from many cross-validation folds) is used in forecasting (it is mentioned test subjects were used, but since there is no test dataset defined a priori or clearly to the reader, it leaves the impression that patients from the same cohort are used in the assessment of models that is not ideal as the gold standard in machine learning literature). The manuscript leaves the impression that there was no a priori, external data set for testing the forecasts. There are some unexpected and odd aspects to reporting results (I have specified below, e.g., using 90% CI), with significant prediction/forecast uncertainty (some of which make the observations less relevant to practice, if at all). My specific comments are below:

- Unclear why confidence intervals are used in a Bayesian paradigm.
- The manuscript should change 90% CI to 95% CI. It seems to the reviewer that if this happens at all (page 6, starting from line 127), the reported biomarker trajectories become non-significant. A major problem in this model is the very wide intervals and uncertainty. It seems that the model has failed to capture variability, and it is unclear why the normal 95% CI convention is changed to 90% CI (this should have been done a priori and with a detailed explanation). The manuscript seems to be selectively reporting 95% CI for other measures.
- The use of the term 'calibration' is vague in Methods. The manuscript probably is referring to some sort of variable adjustment but is not explained. Remove this term and explain specifically what procedure is done to adjust/calibrate models
- Why was the whole brain SUVR used for modeling, while regional atrophy (hippocampus) was used from MRI?
- Choice of linear mixed-effects models and 'no change' models as two benchmarks for the presented model in enrichment is very modest. I suggest using more powerful models designed for non-linear predictions to show to readers how powerful the proposed model is (I can think of deep learning survival models or similar, my point is that the manuscript is not using a challenging benchmark).
- I suggest using the continuous SUVR measures for amyloid-beta, even if different tracers are used. Dichotomizing discards significant useful information in such a rich and large cohort of patients.
- I also find the division of patients into slow/fast progressors problematic and not the best practice. As I mentioned above, using a forecast model, a continuous or probabilistic outcome prediction should be used to enrich the trial. The manuscript seems to be going back and forth between continuous outcomes / mean square error and AUC/ dichotomization (this is very confusing for the reader). Similar to the point above, I believe lots of important data and nuances of modeling are lost because of this thresholding.

Point-by-point response to the reviewer's comments

Reviewer #1:

Q1 *The paper has somewhat of an identity crisis and is split into two halves. The first part is describing the model, while the second is on applying the model and how this may impact trials. The paper is very methods heavy, and as a result isn't ideal for the targeted journal. I felt at times that there simply wasn't enough explanation of what is going on. As a result the paper often had a black box feel to me.*

R1 As explained above, we re-focused the article on the latter part: the application of disease progression models to design more powered trials, not on a detailed explanation of AD Course Map. This is where the core contribution of our work lies. As a consequence, we removed the qualitative description of AD Course Map (previous Figure 2 and associated text), and focus on how to use it as a method for selecting patients in trials. We also included two additional figures: one presenting how the model is used to forecast disease progression (now Figure 1) and another figure to illustrate how these forecasts may be used to decrease sample size in trials (Figure 4).

To limit the impression of using a model (AD Course Map or RNN-AD) as black-box, we added the following description of each method before presenting the results:

AD Course Map assumes that these endpoints follow a logistic progression curve during disease progression with distinct progression rate and age at the inflexion point. It learns how this set of logistic curves need to be adjusted to fit individual data by changing the dynamic of progression and disease presentation (i.e. the relative value of the endpoints at a given disease stage). By contrast, RNN-AD learns how the values of the endpoints will change in the next month given the values of the endpoint at a given time-point. The 1-month transition is assumed to be a non-linear function (e.g. a neural network) of the current value of the endpoints and the current diagnosis.

Q2 1. (a) *The training set uses the 823 amyloid positive ADNI participants. The authors indicate that they used the other ADNI participants as one of the five test sets. It would seem problematic to have an AD prediction model on individuals that you are a priori screening out to not have AD pathology.* (b) *Later on in the methods the authors indicate that they determined the threshold for be the value in the logistic regression that best discriminated amyloid positive ADNI participants from the rest of the cohort. So were the amyloid negative individuals used when determining the curves or not? (a, continued) When hitting the second of the forecasting endpoints it is the held out 20% of amyloid positive individuals and the rest of the cohort that are used. Again, it seems meaningless to predict clinical progression using AD biomarkers in individuals free of such pathology.*

R2 Regarding reviewer's point a: the training set includes only participants who had a confirmed pathological amyloid level at some point during the study. Therefore, the curves (e.g. the model parameters) are determined using only amyloid positive participants. We clarified this in the text: **We train disease progression models using the ADNI participants with confirmed pathological amyloid levels as the training set (N=866) with baseline and all available follow-up data. We kept the data from the other ADNI participants and the members of the four external cohorts as the validation set (N=3,821).**

Once the model is trained, we do apply it on any other participants who might not have amyloid levels above the pathological threshold. They might be participants with unknown CSF amyloid levels, or with amyloid levels increasing over time but remaining below the positivity threshold during the study. Table 1 provides the number of such participants in each cohort. We believe that using these participants in the validation set is legitimate to assess the robustness of the method if it is used on poorly characterized patients.

The forecast errors are corrected for the amyloid status (among other factors including the tau, neurodegeneration and cognitive status). The forecast errors reported in the Figure 2 is for an amyloid and tau positive participant. The change in the error when the method is applied on an amyloid negative participant is shown in the Figure 3, e.g. between 1.8 and 2.5 points decrease in the forecast error of the ADAS-Cog13 by AD Course Map as shown in the last 3 rows of the right panel of the Figure 3 corresponding to suspected non-Alzheimer pathologies. For the detection of the fast progressors and the sample size calculation, the presence of amyloid negative participants depends on the initial selection criteria of the trial. Recent trials testing amyloid antibodies do request amyloid positive participants at entry. So do our simulations.

Regarding reviewer's point b, the threshold mentioned there was to support the description of the pathological cascade found in AD Course Map. This part has been removed in this revised version of the manuscript. It did not play any role in the subsequent analysis.

Q3 2. *The forecasting of data points is interesting, but the paper could do a better job describing that longitudinal data is present. Right now there is just the table. It would be useful to have a bit of this data in the methods text as well (n visits, duration of follow-up). Were all modalities present at all visits, or are some data points only there at a subset of visits (e.g. tau PET was added later in ADNI so you don't have 17 years of follow-up). This would make it easier on the reader to grasp what you are doing without having to jump around. This is a personal preference though more than anything.*

R3 To address the reviewer's concern, we have added the following paragraph at the beginning of the Methods sections:

We used the data from five longitudinal multicenter cohorts: the Alzheimer's disease neuroimaging initiative (ADNI)²⁵⁻³¹ (N=1,652), the Australian imaging, biomarker and lifestyle flagship study of aging (AIBL)^{32,33} (N=460), the Japanese Alzheimer's disease neuroimaging initiative (J-ADNI)^{34,35} (N=470), the PharmaCog cohort^{36,37} (N=111) and the MEMENTO cohort³⁸ (N=1,994). The five cohorts are longitudinal observational studies with an average observation period ranging from 2.0 years for PHARMACOG to 4.8 years for ADNI, with an average number of visits ranging from 3.7 in AIBL to 6.9 in MEMENTO. We considered all participants with at least one year of follow-up. The socio-demographic, genetic, biological and clinical characteristics of the selected participants are reported in Table 1, as well as the proportion of available data in each cohort.

Q4 *There were other parts of this section that just didn't make sense to me. In the methods you state "with a follow-up visit between 1.4 and 6.6 months after the last known visit." Do you mean follow-up after the baseline visit? How can you verify a prediction error in a time point after the last known visit (i.e. no acquired data)?*

R4 We apologize for the typo ("months" instead of "years") and for the term "last known visit", which was misleading – we meant "last unblinded visit". Indeed, for assessing forecast errors, we blinded the latest follow-up data and considered only the baseline and possibly some early follow-up data for predicting the blinded data. We changed the terminology accordingly throughout the manuscript.

Q5 *You then examine the forecasting error using a LME because "multiple forecasts originated from the same participant." Why is this occurring? Is this because you have multiple follow-up visits and you are looking at the error at each one? Since this is the key element of the paper please read through this section and make it as clear as possible.*

R5 The reviewer is right. We assessed multiple forecast errors for a single participant because we have multiple follow-up visits and are looking at the error at each one, and also because we can use different sets of visits to predict the next ones. To clarify this important point, we added the following paragraph in the Methods section:

We used a combinatorial procedure to generate prediction tasks, as described in Supplementary Figure S1. Because we have multiple follow-up visits, we assessed several forecast errors for a single participant: we blinded the data of the participant except at one to three consecutive visits, we predict the individual trajectory using the unblinded data, and forecast the data at the blinded visits after the latest unblinded visit.

Q6 *3. (a) How does the model handle missing data? (b) Was there a threshold of complete data needed for inclusion?*

R6 AD Course Map is generative mixed-effect model. It can be trained and tested with missing data: the likelihood is optimized using the available data only. No imputation was performed. We did not use any threshold of data completion for inclusion in our analyses. By contrast, RNN-AD requires data imputation at baseline. Following the guidelines of the authors of this method, we imputed data using the average endpoint value.

We have added the following sentences to clarify this point in the manuscript: **AD Course Map can be trained and tested with missing data: the likelihood is optimized using the available data only. Model training is robust to missing data⁶, so we did not perform data imputation. By contrast, RNN-AD needs complete data at the baseline visit. We imputed missing data with the mean value of the endpoint in the training set, following authors' recommendations²⁴; missing data at subsequent visits are imputed recurrently using model predictions.**

Q7 *4. The authors indicate that measures were "normalized" between 0 and 1. Typically when I think of normalization I think of a transformation to z-scores. Instead you are using a Box-Cox transformation and then a linear rescaling to make things 0 to 1. This approach assumes that the clinical characteristics of your different cohorts are similar. This is a strong assumption. Did the authors make any attempt to justify this assumption? Was this simply done as a limitation of the progression model?*

R7 In the first version of the manuscript, we used the Box-Cox transform for measurements with skewed distributions, for which z-score transformation is not recommended. Having said that, we changed the harmonization and normalization procedure in this version of the manuscript, following the reviewers' suggestions.

The harmonization procedure is concerned by making sure the data from different cohorts are comparable. For amyloid PET data, we converted data into a centiloid scale using the equations published in the literature. For CSF data, we centered the measurements and reduced their variance to 1 after controlling for age, APOE genotype and clinical dementia rating scale within each cohort (see also R9 below for details).

We added the following paragraph in the Methods section. For CDF data: **We harmonized the measurements to account for the differences in immuno-assays and participants characteristics across cohorts. Within each**

cohort, we regressed each biomarker against age, APOE genotype, and CDR global score with a linear mixed model with random intercept. We then linearly transformed the measurements so that the intercept is 0 and the total variance is 1 for all cohorts. Harmonization equations used are listed in the Supplementary Table S1 for reproducibility purposes.

For amyloid PET data: These PET SUVR values were converted to the centiloid scale (CL)⁵⁹ using equations from the literature⁶⁰ listed in Supplementary Table S6.

Each method, AD Course Map or RNN-AD, relies then on an internal normalization step. For AD Course Map, harmonized amyloid PET data are clipped between 0 and 100 and converted to a (0,1) scale and harmonized CSF, MRI and tau PET data were clipped at the first and last centile, and then linearly mapped to a (0,1) scale. For RNN-AD, all data are transformed using z-scores.

This normalization, however, is only an internal step within each method. The output of the model is always transformed back to its native scale using the inverse transformation, so that the predicted values are comparable with the true, non-normalized data. For instance, predicted hippocampus volume are converted back into the percentage of the total intracranial volume. See the new Figure 1 as an illustration. As a consequence, the forecast errors can be compared across different methods which use different normalization procedures.

We added the following paragraph in the Methods section:

Both models also need an internal step of data normalization. For AD Course Map, cognitive assessments were normalized to a 0 to +1 scale according to the theoretical minimum and maximum values of each assessment, 0 representing the theoretical best value (unaffected participants) and +1 the worst possible value. Harmonized amyloid PET data are clipped between 0 and 100 and converted to a (0,1) scale. MRI, tau PET, and Harmonized CSF data were clipped at the first and last centile, and then linearly mapped to a (0,1) scale. For RNN-AD, normalization consists in a z-score transformation estimated from training data.

Regardless of the normalization procedure, the output of the models are always converted back to the native scale (and unit) of the measurement before being analyzed (see Figure 1). Predicted values are therefore comparable with the true, non-normalized data. Forecast errors can be compared across methods that do not use the same normalization procedure.

Q8 I tend to not like setting the boundaries of all biomarkers to be the same as this is not how the biomarkers actually behave in the disease. The degree of abnormality for say AB42 and amyloid PET are orders of magnitude larger the deviance in hippocampal volume relative to individuals without AD. As a result using such an approach distorts the actual pattern of abnormality. It would be better to use a reference cohort and calculate actual z-scores. In this way you still get values onto the same scale, but you preserve the fact that different markers show variable levels of divergence from normality. This may though be a limitation of your progression model.

R8 The abnormality thresholds in the first version of the manuscript were used only to show that the pathological cascade found by AD Course Map was indeed in line with what has been reported in the literature. It was intended to serve as a qualitative evaluation of the model. We understood that this part created confusion among reviewers. We removed it, as it did not play any role in the subsequent analysis.

Q9 5. Combining CSF data from multiple cohorts is quite difficult. This is due to the wildly different ranges that can be produced across assays as well as the fact that there is substantial drift over time in AB assays (Bijms et al., 2018, Schindler et al., 2018). This drift is likely less of an issue in the ADNI and J-ADNI data but will be an issue for AIBL, PharmaCog, and Memento data. How was variability across assay lots account for?

R9 We agree with the reviewer about the difficulty to compare CSF biomarker values across different immunoassays. Following the reviewer's comment, we revised the way we tackle this problem. We have changed the harmonization procedure as stated now in the Methods section:

We harmonized the measurements to account for the differences in immunoassays and participants characteristics across cohorts. Within each cohort, we regressed each biomarker against age, APOE genotype, and CDR global score with a linear mixed model with random intercept. We then linearly transformed the measurements so that the intercept is 0 and the total variance is 1 for all cohorts. Harmonization equations used are listed in the Supplementary Table S1 for reproducibility purposes.

Below are the distributions of CSF amyloid (left) and tau (right) biomarkers for each cohort superimposed. The results reported in the manuscript have been updated following this new data harmonization method. It has changed the results only marginally.

Q10 6. In the paper the authors indicate they are using a whole-brain SUVR for both AV45 and AV1451 but the methods make it clear that this isn't the case. Update the methods early in the paper to make it clear that you are using tau specific summary.

R10 The reviewer is right. We do not use a whole-brain SUVR but a composite SUVR instead. This procedure is explained in the Methods section in the first paragraph introducing PET data:

For amyloid PET, we used a cortical summary region consisting of the frontal, anterior/posterior cingulate, lateral parietal, and lateral temporal regions; data were normalized with a composite reference region consisting of the whole cerebellum, brainstem/pons, and eroded subcortical white matter^{57,58}

Q11 7. The authors indicate that they cannot combine amyloid PET data between cohorts. There should be Centiloid equations available for both ADNI and AIBL. I'm unsure about Memento. Is PET data not available for J-ADNI or PharmaCog? Is tau PET only coming from ADNI?

R11 We agree with the reviewer and follow the suggestion to use centiloid scale. Nevertheless, conversion equations could be used only for ADNI, and corresponding data are not available for AIBL nor could it be computed. We added the following paragraph in the Methods section:

These PET SUVR values were converted to the centiloid scale (CL)⁵⁹ using equations from the literature⁶⁰ listed in Supplementary Table S1. In the AIBL cohort, the processed amyloid PET SUVR data that correspond to the published centiloid conversion equations were not publicly available. In the MEMENTO cohort, amyloid PET SUVR data are not directly comparable with ADNI data and equations for centiloid conversion were not available. Therefore, we used amyloid PET data on these cohorts only to define the amyloid status of the participants, using pathological thresholds provided by these studies.

Q12 8. *The authors indicate that they calculated two different progression models. When reading the introduction the rationale for this wasn't clear. When I jumped to the methods it appears likely that this is done because PET imaging is not present for all of the cohorts. It was never clear to me why you were running these two parallel models that are near identical. It also isn't clear unless you go down to the methods or table that the five cohorts don't all contain PET.*

R12 The reviewer is right that the rationale for having two AD Course Maps was not clear. Our first intent was to test different scenarios depending on the which endpoints are present in the protocol of the study. This idea created complexity without adding much to the paper. Therefore, following the reviewer's comment, we decided to simplify the presentation in this revised version of the work by using only a single AD Course Map with all possible endpoints. The model needs therefore to deal with a greater amount of missing data. Training and testing were done using only available data for each participant, as in the initial version of the manuscript (see also Q6 and R6 above). We analyzed the effect of missing data on the forecast errors as in the first version of the manuscript. This adjustment changed the results only marginally.

Q13 9. *The raw values for the CSF biomarkers across the cohorts will vary as a function of the platform. You may think about denoting this somehow visually in the table to avoid confusion.*

R13 We harmonized CSF biomarker values in this revised version of this manuscript (see Q9 and R9 above). We now report in the Table 1 the values after the harmonization step and explain it in the caption of the Table.

Q14 10. *In the results the authors say "AD Course Map therefore predicts the subject-specific trajectory of biomarker changes from data collected from the subject." From my reading the Course Map is fit using the aggregate group data. How is it taking into account the subject-specific factors? The authors also indicated that the prediction models are taking into account additional historical data? How is this being done? While the general idea of the model is laid out, there is a fair amount that is coming off to me as opaque. Please make what you are doing as transparent as possible.*

R14 We must acknowledge that the presentation of the training, testing, and validation procedure was not clear. AD Course Map, like RNN-AD, are generative statistical models. In the training phase, the models are trained using repeated data from multiple participants, i.e. the training data set. During this phase, the model parameters are optimized such that the data predicted by the model fit the actual data. In AD Course Map, the model predicts a subject-specific curve, and one compares the actual data with the value of the curve at the age of the participant at the corresponding visit. The general principle is similar for RNN-AD except that the model does not predict a continuous curve, but the changes in the data every month.

In the test phase, one uses the data of a single participant who was not part of the training set. We blind the latest follow-up visits. We use the unblinded data to predict the changes of the data over time and compare the forecast value at the blinded visits with the actual data. Therefore, the forecast may be computed using a variable number of unblinded visits. For trial simulation, however, forecast is computed using data at only a single time-point, thus considered as the baseline visit.

We revised several parts of the manuscript to make this procedure as clear as possible, for instance:

We train disease progression models using the ADNI participants with confirmed pathological amyloid levels as the training set (N=866) with baseline and all available follow-up data.

We repeatedly assessed the errors of AD Course Map and RNN-AD for forecasting cognitive endpoints (ADASCog-13, MMSE and CDR-SB) for participants in the validation set. We blinded the latest visits of the participants and tried to predict them from the unblinded data (see Supplementary Figure S1 and Methods for details of the procedure).

We used the disease progression models to forecast the values of the endpoint at the end of the trial from the baseline data for each participant. The predicted outcome was used as a prognostic score.

We also add the new Figures 1 and 4, which should also help to clarify our analysis plan.

Q15 11. The CSF AB42, ptau181, and amyloid PET values start out very abnormal. This is likely a byproduct of constraining the dataset to amyloid positive individuals? This however does highlight my concern in point #4. In AD the degree of abnormality for amyloid and tau markers far outpaces that of any other biomarker. In your figures though volumetrics, which are far less effected, reach a higher level of abnormality. This is almost certainly due to the fact you are fixing the maximum degree of abnormality across all biomarkers to the same maximal value. As a result your model is distorting the relative degree of abnormality of the different markers.

R15 The reviewer is right that the values of the endpoints, once mapped to the 0 to 1 scale, could not be compared. This is exactly why we introduced biomarker-specific abnormality thresholds to map out the chronological order at which the different measurement deviates from normality. Once again, this illustration and discussion were removed from this revised version of the manuscript.

Q16 12. In Figure 3 the authors indicate that they models are better than alternate models. One of those models is a no-change model while the other constrains longitudinal change to be linear. We know that both of those alternate models are almost certainly incorrect beforehand. We know biomarker and cognitive data change. We also know that they typically behave in a non-linear fashion. These comparisons may be an okay qualitative check for your model, but they are not particularly robust comparisons. I would not overstate this result. The improvement may be less from your particular model and more in the weakness of the assumptions in your alternate.

R16 These two alternatives are not as naïve as they seem to be, as they were shown to be to be good predictor of short-term progression. For instance, in the TADPOLE Challenge the linear mixed models was ranked #2 for the forecast of ADASCog13 and #12 overall as of 2020. Although we all know the disease progresses, making the assumption that the endpoint does not change is a good approximation over short periods of time. This is the case because the disease progresses very slowly and measurements are noisy (raw data do not follow a nice increasing trend, but rather a zig-zag curve). Of course, the further we predict in time, worse is the assumption. The same holds for the linear assumption. We all know that the disease progresses in a non-linear fashion over large periods of time, but mathematics tells us that a linear function does approximate well any non-linear curves over a short period of time.

Having said that, we compared AD Course Map with another non-linear disease progression model: RNN-AD. It is a deep learning method, namely a recurrent neural network, which is a legit choice for modeling progression. We used the model designed by the CBIG research laboratory in Singapore. The model ranked #2 overall in 2020 in the TADPOLE challenge. It represents therefore a fair and challenging alternative to AD Course Map. AD Course Map compares favorably to RNN-AD on all accounts.

Q17 13. I have somewhat of a similar concern with the data presented in Figure 5. I think this is fine descriptive data, but there is no comparative data so it is hard to know what to make of the ROC curves. It is difficult to judge of the prediction rates you are getting are good or bad. The marker of the E4 is helpful, but only one isolated point.

R17 We added the ROC curves for RNN-AD in the plot, so there is at least one comparison curve. We found that RNN-AD never outperforms AD Course Map in the identification of the fast progressors. One should also compare

this ROC curves with the bisector. The bisector represents the current selection practice, meaning that one does not make any difference among participants who fulfill the inclusion criteria. This assumption will lead to the orange curve in the sample size in the Figure 6. Both disease progression models, AD Course Map and RNN-AD, show a significantly better performance in identifying the progressors. To clarify this point, we added the following sentence in the revised version of the manuscript:

We plotted receiver operating characteristics (ROC) curves for the six simulated trials (Figure 5). The area under the ROC curve (AUC) of the six simulated trials fell within the 65%-80% range for AD Course Map and within the 55%-80% range for RNN-AD (see Figure 5).

We compared this prognostic enrichment strategy with two alternative methods: selecting participants at random (bisector of the ROC curve) as currently done in most trials, or selecting participants based on their APOE genotype (gray crosses in Figure 5). All selection methods were significantly better than random selection, meaning that disease progression models succeed in identifying the progressors compared to the current practice that does not make any difference among the participants meeting the inclusion criteria. In all but one case, selections with AD Course Map were significantly better than selection on the basis of APOE genotype. RNN-AD also compares favorably against the two alternatives. Nevertheless, it has significantly worse performance than AD Course Map in two out of six tested scenarios, with a drop of 9% and 14% in the ROC AUC. AD Course Map shows therefore more robust results than RNN-AD when the trial design is varied.

Q18 14. Figures 5 and 6 need better context. A reader is not going to want to jump back and forth between the figure and the table to figure out what the predicted measure is.

R18 We changed the text above the plots in the Figures 5 and 6. They read like: “Participants at risk of AD onset (MMSE).” We hope this change will address the reviewer’s concern.

Q19 15. In the discussion you state In this work “we compared theoretically treated participants with a digital “twin” displaying natural disease progression. This approach makes it possible to generate a synthetic placebo participant exactly matching the treated participants automatically. This application of our method would further limit the number of participants in the placebo arms of trials, provided that a placebo effect is included. ” I wouldn’t say this. Your synthetic placebo is just a model prediction. An error in a model prediction would show up as a “treatment” effect in the absence or any real drug effect. Your comparison is using data from real people, not from digital twins. It is very unlikely that this sort of thing would be acceptable to any regulatory agency in the world. I’d recommend striking this assertion from your discussion

R19 We removed this section in the revised version of the manuscript.

Q20 16. I would not say that your model is better than 56 alternative models. None of that data is presented in the current manuscript

R20 We agree that these data are not presented in the current manuscript, as it was established in prior publications. Nevertheless, we keep this sentence in the Introduction section with the supporting reference as it justifies the choice of AD Course Map as a disease progression model. We use also the results of the TADPOLE challenge to motivate the choice of RNN-AD as an alternative method.

Q21 17. Figure 6 is probably the most important element of your paper. While there is a description of what you are doing it really felt to me very much like a black box. For what is in essence really a methods heavy paper the format you are submitting to leads to a very brief communication of what you are doing. This is unfortunate.

R21 We agree that the presented work results from complex methods. We tried our best to present it in a comprehensive, yet accessible way while satisfying the format of the journal. Finding the right balance is always a difficult task. By contrast, R2 found that “The paper presents a lot of detail in a very comprehensive way” and R3 found that “The manuscript is well written and easy to read”.

In this revision, we removed parts that were not necessary to support the main conclusions and use only one AD Course Map for the sake of simplicity. We also added further details about AD Course Map and RNN-AD. We added two explanatory illustrations to ease the understanding of what we are doing, in particular the new Figure 4 that explains how results in the Figure 6 are obtained. We hope that these changes will improve the reviewer's feeling about the presentation of our work.

Reviewer #2:

Q22 *How were the two sets of end points for AD Course Maps selected (line 362)? I would understand the choice of comparing a PET versus a non-PET model (given that PET is expensive and not widely available yet), but miss an explanation for using CBR-SB in the non-PET but not the PET model.*

R22 This concern concurs with Q12 from R1. Reviewers were right that the rationale for having two AD Course Maps was not clear. Our first intent was to test different scenarios depending on the which acquisitions are present in the protocol of the study. This idea created complexity without adding much to the paper. Therefore, following the reviewer's comment, we decided to simplify the presentation in this revised version of the work by using only a single AD Course Map with all possible endpoints. The model needs therefore to deal with a greater amount of missing data. Training and testing were done using only available data for each participant, as in the initial version of the manuscript (see also Q6 and R6 above). We analyzed the effect of missing data on the forecast errors as in the first version of the manuscript. This adjustment changed the results only marginally.

Q23 *The clinical trial simulation explanation would benefit from a figure visually showing the steps in the simulation and analysis. (Like figure 1, but then for the simulation experiment). Especially the 'hypothetical treatment' gets a bit lost in the current form and took me a while to understand.*

R23 We added a new figure illustrating how forecasting disease progression is used and evaluated in trial simulations. This is the new Figure 4.

Q24 *The explanation of AD Course Map is very brief and cannot be understood with[out] reading a bit of Ref 6. Please expand a bit in this paper.*

R24 We added the following paragraph in the Results section to better introduce what AD Course Map (and now also RNN-AD) does. This explanation is expanded in the Methods section.

AD Course Map assumes that these endpoints follow a logistic progression curve during disease progression with distinct progression rate and age at the inflexion point^{21,22}. It learns how this set of logistic curves need to be adjusted to fit individual data by changing the dynamic of progression and disease presentation (i.e. the relative value of the endpoints at a given disease stage). By contrast, RNN-AD learns how the values of the endpoints will change in the next month given the values of the endpoint at a given time-point. The 1-month transition is assumed to be a non-linear function (e.g. a neural network) of the current value of the endpoints and the current diagnosis.

Q25 *Line 239: The authors [claim] that the method is fair as there are no biases due to sex. Please explain how this is ensured, by using balanced training data?*

R25 This sentence of the discussion refers to the analysis of the forecast errors with respect to different covariates including sex and level of education. To clarify, we edited this sentence in the revised version of the manuscript as:

we show here that AD Course Map provides a fair, robust, and generalizable predictive method. It is fair, in that its predictions are not biased with respect to sex, and are only marginally affected by level of education and the age of the participant. The method is robust to missing CSF or Tau PET biomarkers, but in general better results are achieved when MRI and Amyloid PET data are present. The model was trained on data acquired in North America, but it is readily generalizable to participants from Europe, Asia, and Oceania, with no loss of performance. It performed better at the earliest preclinical stages of the AD continuum than at later disease stages, and is therefore relevant for early-stage interventions.

Q26 Figure 3, CDR-SB: I assume the last box in the last subplot should be red instead of green (PET-model predicting CBR-SB). Please correct or clarify.

R26 It was indeed the correct color. The confusion came from the fact that we could not report forecast errors for CDR-SB for the AD Course Map that did not include CDR-SB as input. Anyway, we have updated and simplified this figure as we are now using a single AD Course Map. There should not be confusion anymore.

Q27 Figure 5: I took me quite some time to fully understand the subplots and their relation to their titles. It would help to include the target variable (MMSE, ADAS, CDR) for each subplot between brackets. And to change the second sentence to something like "Receiver operating characteristic (ROC) curves are shown for the six simulated trials showing the performance of AD course map in selecting the group of participants with the largest change in primary outcome during follow-up."

R27 We added the target outcome between brackets above each sub-plots, and changed the sentence in the caption following the reviewer's suggestions.

Reviewer #3:

Q27 My biggest concern about this study is the lack of a real external cohort. While there are more than five cohorts, none is set outside of modeling and training, and as explained in Methods (although difficult to understand), it seems that only N-fold cross-validation is used there. Specifically, in the sample size and power analysis (enrichment part, end of Methods): it is unclear what model (from many cross-validation folds) is used in forecasting (it is mentioned test subjects were used, but since there is no test dataset defined a priori or clearly to the reader, it leaves the impression that patients from the same cohort are used in the assessment of models that is not ideal as the gold standard in machine learning literature). The manuscript leaves the impression that there was no a priori, external data set for testing the forecasts.

R27 It is very unfortunate that we did not clearly explain our validation procedure. In fact, the training data sets only include amyloid positive participants from ADNI. The other ADNI participants and the participants from the 4 other cohorts were never used as training data, only as a validation dataset. There is no cross-validation needed as neither AD Course Map nor RNN-AD require model selection or hyperparameter tuning (for RNN-AD we used hyperparameters advised by the authors). We only used 80% of the training set, so that we can leave 20% of it as a test set. This test set is not used for training or parameter tuning. It has the same status as the external validation set. We did our best efforts to clarify this point in the text with the help of the Supplementary Figure S1.

The section about the validation procedure in the Methods section now reads:

We split the data sets in two (see Supplementary Figure S1). We first considered the ADNI participants who were amyloid-positive according to CSF or PET data on at least one visit (shown in red in the Supplementary Figure S1). We then kept the other ADNI participants and all participants from the four other cohorts as an external validation set (shown in in blue in the Supplementary Figure S1).

We then split the amyloid positive ADNI participants in 5 random folds and trained AD Course Map and RNN-AD using all available data of the participants in 4 out of the 5 folds, e.g. the training set. We repeated this procedure with another split, so that we ended up with 10 instances of each model. Each participant has been counted twice as a test subject in the left-out fold. Therefore, it can be used twice for evaluating prediction tasks with two different instances of each model. By contrast, each participant in the external validation set can be tested with 10 different instances of each model. In the following, we averaged the prediction made by the 2 instances of the model for the participants in the test sets, and by the 10 instances for the participants in the external validation set.

The test subjects did not contribute to any model selection or hyperparameter tuning neither for AD Course Map nor for RNN-AD. Therefore, we pooled the forecasts of test subjects with the ones in the external validation set.

See also the answer R14 to the question Q14 above.

Q28 *There are some unexpected and odd aspects to reporting results (I have specified below, e.g., using 90% CI), with significant prediction/forecast uncertainty (some of which make the observations less relevant to practice, if at all).*

R28 All confidence intervals are set to 95% in this revised version of the manuscript. The 90% confidence intervals were used in the original submission for the computation of the abnormality thresholds in the illustration of AD Course Map. This part has been removed from the manuscript (see e.g. Q8 and Q15). We used empirical 80% confidence intervals when reporting goodness-of-fit residuals (Supplementary Table S1) since we only trained 10 models but they do not play any role in the subsequent analysis. Although differences in forecast errors are small across methods, and often not significant, methods still significantly differ in the detection of the fast progressors and the amount of sample size reduction.

Q29 *Unclear why confidence intervals are used in a Bayesian paradigm.*

R29 We could derive credible intervals for the fixed effects of the Bayesian model. However, they could not be computed for forecast errors, ROC curves and sample size. We resorted to point estimates and resampling methods for computing confidence intervals for these metrics. These intervals can then be compared with the ones resulting from the alternative non-Bayesian methods.

Q30 *The manuscript should change 90% CI to 95% CI. It seems to the reviewer that if this happens at all (page 6, starting from line 127), the reported biomarker trajectories become non-significant. A major problem in this model is the very wide intervals and uncertainty. It seems that the model has failed to capture variability, and it is unclear why the normal 95% CI convention is changed to 90% CI (this should have been done a priori and with a detailed explanation). The manuscript seems to be selectively reporting 95% CI for other measures.*

R30 We changed the 90% confidence intervals for the 95% throughout. 90% intervals were present in the part of the manuscript that has been removed. See R28 above. The uncertainty of the estimated logistic curves of AD Course Map did not play any role in the subsequent analysis, and therefore does not call into question the main results of the article.

Q31 *The use of the term 'calibration' is vague in Methods. The manuscript probably is referring to some sort of variable adjustment but is not explained. Remove this term and explain specifically what procedure is done to adjust/calibrate models*

R31 We agree that the term calibration was misleading. In fact, we meant “training”. We did our best efforts to describe our validation more clearly, following the reviewer’s comments. See Q27 above. We also made sure to adopt a consistent terminology throughout the paper. We hope this will clarify the presentation.

Q32 *Why was the whole brain SUVR used for modeling, while regional atrophy (hippocampus) was used from MRI?*

R32 We used a composite score of regional SUVR, which weights the contributions of different cortical regions as validated and published in the literature (references 57-59). We therefore follow the current standard practice.

Q33 *Choice of linear mixed-effects models and 'no change' models as two benchmarks for the presented model in enrichment is very modest. I suggest using more powerful models designed for non-linear predictions to show to readers how powerful the proposed model is (I can think of deep learning survival models or similar, my point is that the manuscript is not using a challenging benchmark).*

R33 This comment concurs with Q16 from R1. These two alternatives are not as naïve as they seem to be, as they were shown to be to be good predictor of short-term progression. For instance, in the TADPOLE Challenge the linear mixed models was ranked #2 for the forecast of ADASCog13 and #12 overall as of 2020. Although we all know the disease progresses, making the assumption that the endpoint does not change is a good approximation over short periods of time. This is the case because the disease progresses very slowly and measurements are noisy (raw data do not follow a nice increasing trend, but rather a zig-zag curve). Of course, the further we predict in

time, worse is the assumption. The same holds for the linear assumption. We all know that the disease progresses in a non-linear fashion over large periods of time, but mathematics tells us that a linear function does approximate well any non-linear curves over a short period of time.

Having said that, we compared AD Course Map with another non-linear disease progression model: RNN-AD. It is a deep learning method, namely a recurrent neural network, which is a legit choice for modeling progression. We used the model designed by the CBIG research laboratory in Singapore. The model ranked #2 overall in 2020 in the TADPOLE challenge. It represents therefore a fair and challenging alternative to AD Course Map. AD Course Map compares favorably to RNN-AD on all accounts.

Q34 *I suggest using the continuous SUVR measures for amyloid-beta, even if different tracers are used. Dichotomizing discards significant useful information in such a rich and large cohort of patients.*

R34 In this revised version of the manuscript, we used the continuous composite SUVR from two radiotracers converted into a centiloid scale using conversion equations from the literature (see R11 above). This measure was only available in ADNI, though.

Q35 *I also find the division of patients into slow/fast progressors problematic and not the best practice. As I mentioned above, using a forecast model, a continuous or probabilistic outcome prediction should be used to enrich the trial. The manuscript seems to be going back and forth between continuous outcomes / mean square error and AUC/ dichotomization (this is very confusing for the reader). Similar to the point above, I believe lots of important data and nuances of modeling are lost because of this thresholding.*

R35 The considered disease progression models, whether AD Course Map or RNN-AD, predict the value of the endpoints at a given time-point in the future. This predicted value is continuous. It is used to derive a continuous prognosis score, namely the relative amount of change between baseline and the predicted follow-up value. Participants are ranked therefore according to their prognosis score, and this is how the ROC curves are constructed in Figure 5. At this point, to simulate clinical trials, we need a decision rule (i.e. to select a point on the ROC curve) to decide whether a participant has a prognosis score high enough to be included in the trial. Thresholding only occurs at this latest stage of the analysis and is then inevitable. We added a new explanatory figure to better illustrate this procedure. It is now the Figure 4.

We believe that the feeling of going back and forth between continuous and discrete value comes from the illustration of AD Course Maps with abnormality thresholds in the previous Figure 2. With this illustration, we intended to show that the pathological cascade learnt by AD Course Map was consistent with the current knowledge about the disease. It was not used in the rest of the analysis. We hope that the removal of this part will make the reading of the manuscript more straightforward and logical.

Sincerely yours,

Stanley Durrleman, on behalf of the authors.

Reviewer #1 (Remarks to the Author):

The authors have made substantial revisions to the manuscript in accordance to questions from all of the reviewers. I think the paper has improved as a result.

Reviewer #2 (Remarks to the Author):

The authors performed a very careful and rigorous review of the manuscript. I appreciate the addition of another method for comparison (RNN-AD of an external research group), and the changes in the experiments made to improve clarity. All my comments are addressed successfully.

Reviewer #3 (Remarks to the Author):

All my comments have been addressed with significant extra work from the authors.

Arman Eshaghi